

# Quantitative analysis of actors' mention in press coverage of a seismo-volcanic activity in the French overseas

Louise Le Vagueresse[1], Marion Le Texier[2], Maud H. Devès[1,3]

[1] Université Paris Cité, Institut de physique du globe de Paris, CNRS, F-75005 Paris, France

[2] Université Paris Cité, Centre de Recherche Psychanalyse Médecine et Société, CNRS, F-75006 Paris, France

[3] LAGAM, Laboratory for Geography and Territorial Planning in Montpellier, University Paul Valéry, Montpellier 3, France

*Correspondence to:* Louise Le Vagueresse (levagueresse@ipgp.fr)

**Abstract**

Media, especially the press, play a crucial role in shaping public understanding and representations during risk and crisis management, acting as intermediaries between various actors and the public. However, their framing of sources can introduce biases into representations. Limited analysis exists regarding how press coverage portrays relationships between crisis and risk management actors. Using Social Network Analysis, we map quotation networks in press coverage of a seismo-volcanic crisis in Mayotte, a French overseas department allowing us to: i) have an overview of the relationships between actors; ii) highlight unique aspects related to the context and media portrayal; iii) display underlying representations and levels of trust among interviewed actors and iv) visualises networks' dynamics over time. Analysis revealed variations in narrative approaches among newspapers, with some focusing on specific aspects. General results show that national authorities received more attention than local elected representatives, and scientific figures dominated reported speeches, while the population's perspective remained relatively passive despite their centrality to the quotation network. Identified individuals held significant positions, emphasising the importance of personal connection in communication and revealing a potential distrust toward political and scientific institutions. This underscores the need for proximity between sources and the community.

## 1. Introduction

Risk communication is a key component of disaster risk reduction (UNISDR, 2015). Implementing an efficient risk communication strategy is however not a trivial matter (e.g. Drabek, 1986; Mileti and Sorensen, 1990; Tierney, Lindell and Perry, 2001). There are many pitfalls: in the communication process between actors in charge of risk monitoring and management as well as in the process of public information sharing. It is particularly difficult when uncertainties are large, which is the case in crises related to volcanic hazards for instance (e.g. Barclay et al., 2008; Solana et al., 2017; Andreastuti et al., 2019). Mass media (newspapers, television, radio) play an important role with regards to public information (see Perry and Lindell 1989 p. 47-62 or Scanlon, 2007 for an overview). In crisis situations, they are identified as the main source of information for the public while searching for hazard-related information (Nazari et al., 2011; Poudel et al., 2015 ; Van Belle, 2015). It is especially the case for local and national media (Burkhart, 1991; Allan et al., 2000 ; Scanlon, 2007) and their participation is thus crucial for effective warning (Lindell et al., 2006). News reports are also closely followed by crisis management teams influencing official communication strategies (e.g.,



Lagadec, 1991; Lindell et al., 2006). They affect risk perception in the long term, notably by contributing to the
circulation of "erroneous representations" about how individuals, groups or organisations behave during disasters
(Coleman, 1993; Quarantelli, 2002; Wachinger et al., 2013; Van Belle, 2015).

As compared to other sources, newspapers, especially the daily press, are commonly seen as a more credible source of
information because of their ability to provide in-depth analytic coverage (Quarantelli 2002, cited by Steelman et al.
2015). They are also widely relayed in other media or on social networks. The local press occupies a specific position to
this respect, as local journalists are both interested parties and commentators of ongoing crises. The resulting coverage
tends to be more regular and more detailed and it often provides the raw material for press agencies and, through them,
for the other media (e.g. Nielsen, 2015 on how local newspapers act as a "keystone media" despite having few readers
and Cagé, Hervé and Viard, 2017). Studies have also demonstrated the pivotal role played by local journalists as
intermediaries between risk management authorities and populations while disseminating warning messages, conveying
the community's concerns and providing updates on the situation at the grassroots level (Scanlon, 2007). Newspapers'
coverage constitutes therefore an important issue for disaster research (see Harris et al. 2012; Camilleri et al., 2020;
Calabrò et al., 2020;  Le Texier et al., 2016 and Devès et al., 2019 for application to seismic crises and Devès et al.,
2022b for applications to volcanic crises).

However, the daily press, and the media in general, cannot be considered as simple vehicles for providing information. As
recalled by Aylesworth-Spink (2017), they act as "a complex mediator with specific interests and motivations". The way
the media depict an event is neither exhaustive nor neutral. There are many factors influencing the final coverage:
selection of topics (Pavelka, 2014), layout and design choices (e.g. Moirand, 2006; Schindler and Krämer, 2017; Billard
and Moran, 2023), political ideology and editorial policy of the newspaper (e.g. Wang et al., 1992; Shoemaker and Reese,
1996), access to sources and their respective social status (Ploughman, 1997), choices of contextualization (Llasat et al.,
2009; Cavaca et al., 2016; Carter et al. (2018) in the context of Christchurch 2010 and 2011 and Kaikōura 2016
earthquakes). Day-to-day journalistic practices also play a role (e.g. Boykoff and Boykoff (2004) about quoting sources
on an equal footing on the example of global warming). The way journalists tend to cross the speeches of heterogeneous
sources, whether important for depicting the variety of viewpoints, has been shown to "blur" messages (e.g. Lejeune,
2005, Léglise and Garric, 2012, and Devès et al., 2022a). These various factors can lead to conveying representations to
the public that are sometimes very different from how authorities and scientists see the situation (Ploughman, 1995).
They may also implicitly replicate common misconceptions (see Quarantelli, 1996) or reproduce asymmetrical power
relationships between actors without really questioning them (local vs national authorities, experts vs lay public, etc., see
Valencio and Valencio (2018) on the under-representation of at-risk communities' vision about recovery solutions for
their lives or Devès et al. (2023) on reproduction of asymmetrical power relationships in the media discourses in the
context of a French oversea seismo-volcanic crisis).

Examining press coverage provides insights into the pivotal moments and key actors perceived by journalists covering the
event, who often serve as primary observers on the scene. The use of content and thematic analyses allows for the
reconstruction of the sequence of events, mapping of actors' networks (Hijmans, 1996), and identification of
representations conveyed by the press toward at-risk communities. This is exemplified by studies like those conducted by
Thistlethwaite & Henstra (2019) or Calabro et al. (2020). However, there is a limited number of studies analysing how



relationships between actors are portrayed in the press. Do these networks of interrelations, commonly connecting crisis
management actors, experts, and populations, align with the envisaged distribution of roles during crisis management
planning? If disparities exist, what insights do they provide?

Examining such interrelations can be accomplished by creating maps of quotation networks, representing how the actors
themselves cite or reference each other in the press text (McLaren and Bruner, 2022). This method falls within Social
Network Analysis (SNA), a widely employed approach in social and information sciences (Otte and Rousseau, 2002;
Sapountzi et al., 2018). SNA utilises tools from network analysis and graph theory to investigate social structure and
information circulation within networks of actors. Past studies utilising SNA or its derivatives in the realm of disaster risk
research have shown interest in examining misinformation and the structuring of information networks in social media
(e.g., Pourebrahim et al., 2015; Kim et al., 2018), or conducting a functional analysis of crisis management organisations
(e.g., refer to Trias et al., 2019, for governance, and Flecha et al., 2023, for humanitarian aid). To our knowledge, there is
a lack of studies utilising SNA on press data within the context of risk or crisis management. Yet, this is an important area
of research, as this approach enables: i) gaining insights into the actual organisation of actors by providing a
comprehensive view of all cited actors and their interactions, allowing the detection of communities (e.g. Park et al., 2015
and Williams et al., 2015); ii) identifying actors who are prominently featured, whether due to their perceived reliability,
relevance in transmitting information on a subject, specific social role or accessibility to journalists, iii) examining the
involvement of various actors and the evolution of this network over time (including the appearance and disappearance of
actors and its effect on the structure of the network) and, iv) accessing a particular representation of actors, whether active
or passive in media coverage. Here, we apply this method on a press coverage of a seismo-volcanic crisis in a French
oversea department, Mayotte.

Before presenting our corpus (Section 3.1) and methods (Section 3.2), we briefly describe Mayotte geological and
sociological contexts and explain why it is an interesting case study (Section 2). We then expose our findings (Section 4).
Section 4.1 concentrates on the actors' mention frequency and form in varying newspapers (depending on publication
rate). Section 4.2 focusses on whether actors' statements are displayed directly or through a third party, depending on the
newspaper. Section 4.3 explores the positions of the actors mentioned in the chain of quotation. Section 4.4 is a
comparison of the actor network structures during several specific "moments" of this media coverage. In sections 5.1 and
5.2, we discuss differences between  the press representation of the actors network and the official organisation of risk
and crisis management in France and its overseas territories. Eventually, we conclude on the interest and caveat of our
approach and on future avenues of research.

**2. Case study description**
Devès et al. (2022a) provide a detailed description of Mayotte's geological context and the so-called "seismo-volcanic
crisis" that began in Mayotte in May 2018. We settle here for reminding the main events and reviewing the latest
scientific updates since knowledge evolves quite rapidly in the area due to ongoing significant research efforts.

The seismo-volcanic crisis began on the 10th of May 2018 with an unusual seismic activity (tens of felt earthquakes in the
first month alone, with magnitude up to $M_w$ 5.9, see BSCF 2018). This seismic crisis turned out to be linked to a volcanic



eruptive activity at sea and a newly born volcanic edifice, named Fani Maore, was discovered one year later, in May
2019, at about 50 km off the eastern coast of Mayotte islands. From a scientific perspective, uncertainties were really
high, especially in the first months of the seismic crisis due to scarce knowledge of the geodynamical context in the area
and a poor instrumental network (Saurel et al., 2021; Bertil et al., 2021; Feuillet et al., 2021). This made public
communication particularly difficult and led to the development of a "technicalist bias" with frequent, but technicalist and
minimalist communication from institutions that did little to help the population to appraise the situation (Devès et al.,
2022a). Indeed, there was an overall feeling of "lack of information" (Fallou et al., 2020) which led to the spread of
numerous rumours to explain this phenomenon and to regular complaints from inhabitants and their representatives (see
for instance the questions addressed to the government by a Member of Parliament for Mayotte Ali in 2018, as well as the
opened letter sent to the authorities and scientists by a group of citizens in February 2019, Picard, 2019). At the time of
writing, the eruption of the new submarine volcano Fani Maore has ceased and seismic activity is divided in two main
clusters active since the end of June 2018 (according to Lemoine et al., 2020) at respectively 5 to 15 km and 30 to 40 km
from coast (Feuillet et al., 2021; Saurel et al., 2021; Lavayssière et al., 2021). Most of them are volcano-tectonic seisms.
Another sign of activity is the detection at 10 to 15 km from the coast of acoustic plumes associated with geochemical
anomalies (22 sites observed in July 2022, MAYOBS 23) and possibly linked to the gas emissions monitored onland on
Petite Terre island since prior to 2018. Hence, magmatic processes related to these observables are still pretty close to the
island. As their uncertain evolution presents a significant hazard, it is currently being monitored by REVOSIMA
(Mayotte Volcanological and Seismological Monitoring Network).
In addition to scientific uncertainties and the already mentioned ensuing difficulties in public information, other factors
could also have undermined the relationship between actors. For instance, there is both a geographical and a cultural gap
between scientists involved and the local populations since most of the former ones are based either in mainland France
or in La Réunion island and do not have lots of occasions to exchange with Mayotte inhabitants. As detailed in Devès et
al. (2022a), Mayotte is a multicultural archipelago with a dominant oral culture where about 37% of the population do not
speak French (INSEE, 2017), which complicates risk prevention communication from scientific institutions and
authorities in charge. It is also a particularly vulnerable territory marked by poverty and important social inequality
(Roinsard, 2014; INSEE, 2021). Since its recent departmentalization in 2011, it has been regularly shaken by social crisis,
one ending just as the seismic swarm began (Roinsard, 2019; Mori, 2021). Finally, there seemed to be no living memory
of seismic and volcanic phenomena in Mayotte implanted in the population. The last important earthquake was a
magnitude $M_L$ 5.3 in 1993 (Bertil et al., 2021). This added to the underwater nature of this activity brought people to
confusion, some of them going so far as to doubt the scientific explanations and the very existence of a volcano, even
today (see testimonials in Devès et al., 2023). In this context, which brings together strong scientific uncertainties, social
tensions and a multitude of players, we are seeking to identify in greater detail the obstacles and mechanisms that have
hampered the information at each link of the communication chain.
To sum up, Mayotte's seismo-volcanic activity is an interesting case for this study because: i) although the seismic-
volcanic phenomenon itself has been associated with moderate impacts, in the first years of activity, it triggered a social
crisis that the risks managers themselves qualified as a "communication crisis" (see questions to the government, Ali
(2018) where the deputy Ramlati Ali expresses in national assembly a need for information and an open letter sent by a
citizens' group (Picard, 2019) and I which state services, elected officials and scientists were taken to task on this subject.



More details are exposed in Devès et al. 2022a and b, Fallou et al. 2020, Mori 2021 and 2022); ii) despite a large quantity
of public information documents issued by scientists and the authorities (Devès et al., 2022a), the significant feeling of
"lack of information" within the exposed population documented by Fallou et al., 2020 raises questions about the
transmission chain of this information to the public; iii) risks are perceived mostly indirectly by at-risk populations, which
poses specific challenges for public information (Skotnes, Hansen and Krovel, 2021); iv) there are large uncertainties,
some of them still ongoing as we write; and eventually v) the activity is long-lasting allowing to study the evolution of a
large coverage over time.

**3. Method**
**3.1 Corpus**

We build on from two previous studies (Devès et al., 2022a which focuses on public information processes and showed
caveats in both scientific and state institutions communication and Devès et al., 2023 which illustrate how newspapers
implicitly reproduce asymmetrical power relationships between actors without really questioning them (local vs national
authorities, experts vs lay public, etc)) to identify and compare occurrences of actors according to their role in the risk
reduction network, the geographical scope of newspapers (local vs regional vs national) and whether there are significant
differences between newspapers. We use the same corpus as Devès et al. (2022b) and Devès et al. (2023) that contain
articles from six French-written daily newspapers published between the 10th of May 2018 and the 10th of May 2021.
The methodology for creating this corpus is inspired by Le Texier et al. (2016) and Devès et al. (2019). We selected
newspapers based on four criterias: 1) number of articles published on our case study, 2) broadness of readership, 3)
spatial distribution of the readership along with the structural and cultural links existing between this zone and the study
case area and 4) a targeted language (French in this study). Six French language daily newspapers were selected among
56 sources mentioning these events, addressing national (*Le Monde, Le Figaro*), regional (*L'Express de Madagascar, Le*
*Journal de l'île de La Réunion*) and local readerships (*Le Journal de Mayotte and Mayotte la 1ère*); see Figure 1 and Part
1 in Supplementary Information. Articles were then collected with two types of sources : 1) Factiva and Europresse, two
full-text press databases offering a selection of general and specialised of both paywalled or freely accessible newspapers
with regional, national and international readerships, and 2) web archives, especially for local press articles which can not
be found in those two databases. Factiva and Europresse databases are often used by scholars to study media coverage
(Severo et al., 2015, Reboul-Touré 2021, Bernier et al. 2013).
The resulting database is composed of 358 articles published between May 10th 2018 and May 10st 202 thus covering the
first three years of Mayotte's seismic-volcanic crisis. We chose to limit this database to May 2021, after the pictures
taken underwater and graphical representations of the phenomenon began to be regularly broadcasted by these media thus
involving possible changes in the perception of the phenomenon by the readership. It is also when articles in the local
press become less frequent. Within this database, 15 articles were excluded (see Supplementary Information) in this
particular study as we wanted to work only on news items whose main subject is the seismo-volcanic activity. Therefore,
the final database includes 343 articles and covers the first three years of the recent seismo-volcanic unrest near the
archipelago of Mayotte.



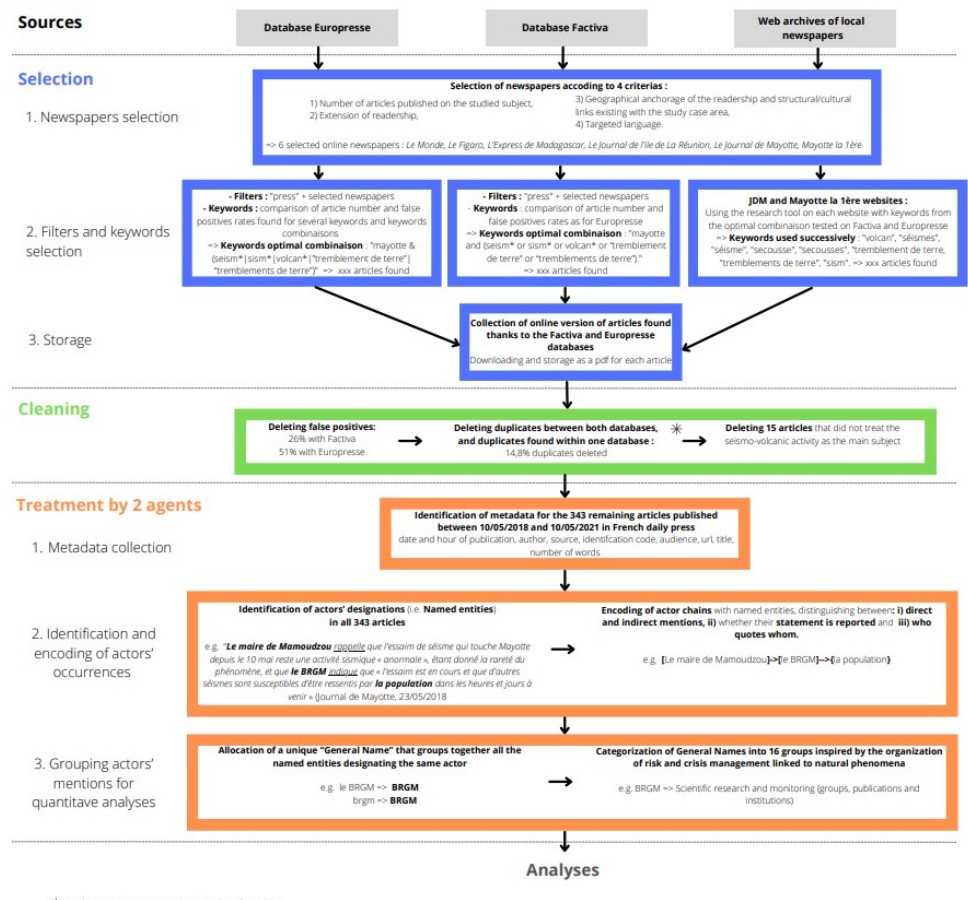


**Figure 1: Schematic view of the process of newspapers' articles collection in treatment before analyses. The articles are selected from 3 main sources using a combination of keywords (see Supplementary Information for more details on the keyword analysis). Articles are all read a first time in order to identify and delete false positives and duplicates. False positive and negative rates are determined on representative samples (see Supplementary Information). Each article is eventually read independently by 2 to 3 researchers who complete a data table with metadata and treatment variables. Disagreements were discussed and solved collectively.**

## 3.2 Indicators

### 3.2.1 Actors and categories of actors

We use a broad definition of the term "actor" which encompasses both individual and legal entities, groups of individuals sharing a common character or purpose such as a scientific mission or being impacted in the same way by the crisis, but also media agencies who play an active role in communicating about the event (Agence France Presse, Twitter, scientific journals, etc.). Places or buildings (mobile like a boat or immobile like a school) may be described in the media as actors when they are named as synonymous for the individuals they host and are then also selected.



To study the press coverage of different categories of "actors" in our corpus, we first followed a double-reading method
by human operators enabling us to select and identify each actor or group of actors mentioned in articles, even when they
were identified by professional status, by nicknames, etc. This qualitative analysis of the texts in the corpus made it
possible to disambiguate the majority of references to actors in the texts. For example, the terminology "experts" can be
used to refer to scientists specialising in the hazard as well as to technicians from the Bureau Veritas in charge of
assessing the damage and some actors have different affiliations depending the articles due to errors or evolutions in
his/her career (e.g. Nathalie Feuillet, a researcher, has been wrongly affiliated to IFREMER in some articles like for
instance 20190507_JDM_001). A careful reading of the articles is thus needed and generally allows the actors to be
categorised. Remaining actors are labelled as "unidentified" in the category Divers/Unidentified (see Part 2 in the
Supplementary Information). When an actor is identified in the press article, we note its exact denomination(s) (i.e.
named entities) and we build two correspondence tables allowing to : i) identify the different ways of naming the same
actor and grouping them under a chosen "general name" (TABLE NamedEntitiesToGeneralName in TABLE SA) and ii)
group these actors in categories (see TABLE GeneralNamesToCategories in TABLE SB for a correspondence table
between "general names" and "categories") in order to build a structural analysis.

**TABLE 1: Denominations and definitions of categories used to group actors identified by two human operators in a database of**
**343 daily press articles published between 10/05/2018 and 10/05/2021 covering the first three years of the recent seismo-**
**volcanic unrest in Mayotte archipelago. Categories are determined according to the organisation of risks and crisis**
**management in France (see Fearnley et al. (2018) and Section 2 in Part 2 of the Supplementary Information).**

| Level 1 of categorization: | Usual categories considered in risks and crisis management studies |
|---|---|
| Name of categories | Definition of categories |
| Scientific research and monitoring (groups, publications and institutions) | Scientific groups, publications, institutions and all groups of people involved in monitoring and research on the sismo-volcanic activity in Mayotte. |
| Scientific research and monitoring (named individuals) | Namely identified scientists involved in monitoring and research on the sismo-volcanic activity in Mayotte. |
| Risks and crisis management actors | Administrative authorities involved in risk and crisis management activities |
| French political institutions | French political institutions involving members of the government, the French Parliament and the Senate |
| Public and para-public services to the population (institutions and members) | French public or parapublic services to the population |
| Elected local officials | Locally elected executive representatives |
| Mass media and associated journalists | Includes TV, radio, magazines, newspapers and associated journalists |
| Social media/Internet | Social media or websites |
| Civil society, private sector and NGOs | Civil society, private sector and NGOs |





| | |
|---|---|
| Local identified personalities | Influential figures in Mayotte |
| At-risk populations in Mayotte | Populations living in Mayotte and exposed to natural hazards |
| Educational staff and institutions | Educational staff and institutions in Mayotte and mainland France |
| Students and schoolers in Mayotte | Children living in Mayotte when mentioned in school contexts |
| Other populations | Populations living outside of Mayotte |
| Foreign states, communities and personalities | Foreign state actors, personalities or communities that are not involved in scientific or risk and crisis management activities |
| Divers/Unidentified | All actors that could not be categorised in the previous categories because unidentified, or belonging to more than one category |


### 3.2.2 Direct vs indirect mentions and reported speech vs simple mention
A direct mention is when an actor is cited in the text body by the author of the article without being introduced by another
actor. Indirect mention corresponds to the case when an actor is mentioned through a third party in the article. For
example, in the sentence "An earthquake with a magnitude of 4.0 was recorded by the Bureau of Geological and Mining
Research (BRGM) informs the prefecture", the citation of the prefecture is declared as direct and that from the BRGM as
indirect. This distinction allows us to measure the interactions between categories of actors in the press, and in particular
the most frequent citation links, the direction of these relationships (and therefore their potential asymmetry) and finally
the more or less central position of actors and categories of actors within the citation network.
We also draw a distinction between reported speech and simple mentions of actors (see TABLE 2). Reported speech can
be direct or indirect. What we identify as reported speech includes everything that the journalist presents as being the
word or opinion of this actor, whether it appears to be reported directly (with the use of quotation marks for example) or
indirectly, or even distorted. For e.g. in the sentence : "In May 2018, when the swarm of earthquakes began to shake
Mayotte, the first scientists rushing to the island did not believe in volcanic activity", we consider that a voice is given to
the scientists since the news item is supposed to convey their beliefs. On the contrary, in the sentence "End of mission:
French prefect Dominique Sorain leaves Mayotte", we consider that Dominique Sorain is "simply mentioned".



**TABLE 2 : Illustration of the distinctions direct mention vs indirect mention and reported speech vs simple mention using the actor "prefecture".**

|  | Direct mention | Indirect mention |
|---|---|---|
| **Reported speech** | "The **prefecture** *confirms* that no fewer than 13 tremors were recorded." <br><br> Journal de Mayotte, 05/12/2018 | "On Tuesday evening, **they [prefecture]** had *to deny* on **Twitter** a rumor indicating that a strong magnitude earthquake could occur soon '*This rumor is totally unfounded*'. " <br><br> Le Figaro, 05/16/2018 |
| **Simple mention** | "The Mayotte **prefecture** activated its crisis unit this morning." <br><br> Le Figaro, 05/16/2018 | "**They [STTM members]** criticized the **prefecture**'s poor communication." <br><br> Mayotte la 1ère, 08/08/2019 |

### 3.2.3 Actor network analysis

We describe the system of actors in the Mayotte seismo-volcanic activity reported in the press using two global network analysis indicators and further detect the presence of small communities using the Louvain clustering method. We study the system of actors depicted by the network of citations in order to better understand the relationships between people and categories of actors and their evolutions using network centrality indices and network diagrams plotting citation links with arrows and weighting the size of the nodes and of the fonts of the generic names by their number of connections (degree). Unidentified actors are removed from the graph plot to avoid false co-citation relationship structures.

In order to study the position of the actors in the citation networks (source and destination of a citation) derived from the corpus, we use two indicators from network analysis at the node level: degree centrality and betweenness centrality. Degree centrality measures the number of links held by a node. It captures the amplitude of the network with which an actor is connected in the media, through the citation process. We distinguish in- and out-degree centrality, i.e. the number of actors by which an actor has been mentioned and the number of actors that he/she has mentioned. The actors with the highest out-degree index values are those with the strongest activity in transmitting and communicating the experiences, opinions, speeches and actions of other stakeholders (including Mayotte's population). On the contrary, a high in-degree index demonstrates a central position in the network linked to strong interest from third parties. The study of the ratio between in and out-degree centrality makes it possible to study the level of reciprocity of these two states. Betweenness centrality measures the number of times a node lies on the shortest path between two other nodes. The values are normalised by the number of node pairs in the graph (direction of citations are not accounted for). A high betweenness index indicates that an actor plays an important role in connecting the network of actors depicted in the media, and in particular the subgroups that the citation relationships update, either because he/she positions himself/herself at the centre of the network, or because he/she is positioned on the periphery of several clusters. Actors with high betweenness are key bridges between different parts of a network.

Here, we provide 8 different graphs matching the 8 major periods subdivision of the seismo-volcanic activity press coverage proposed by Devès et al. (2022b). This allows us to examine the evolution of the network at different periods, each characterised by the occurrence of a new external disturbance (first earthquakes, first discussions regarding the hypothesis of a volcanic origin, discovery of the volcano, public conference, etc).



282

## 4. Results

### 4.1 Actors' mentions in the corpus

TABLE 3 presents a set of descriptive statistics on the frequency and form of actors' mentions in our corpus. The newspapers under study have not all published with the same frequency on the events: the national daily *Le Monde* devoted 10 articles to the subject, while 190 articles were published by the local daily *Journal de Mayotte*. Differing publication rates result in differing actors' mentions rates depending on newspapers: *Le Monde* names 310 actors (often repeatedly), while this number peaks at 2,541 for the *Journal de Mayotte*. Beyond this rate effect, we also find differences in the diversity of the actors that are mentioned: *Le Monde*, the national daily *Le Figaro* and the regional daily *Le Journal de l'Île de la Réunion* respectively mentioned on average 31, 20 and 20 actors per news item, while this mean value only reached 13 for *Le Journal de Mayotte*, 12 for the local daily *Mayotte la 1ère* and fell to 10 for the regional daily *L'Express de Madagascar*. An inverse relationship emerges between publication rates and the average number of actors' mentions per article. Ultimately, the position of actors in the citation network will be driven by the most prolific media, which we control by building our analyses on relative indicators.

Another source of variability in the media coverage of the actors lies in the space left by each newspaper to the direct or indirect mention of actors, and the reported speech of these actors versus their simple mention. If the trend is +/-80% for direct mentions per article (compared to +/-20% for indirect mentions), this proportion drops to 66.7% for the national daily *Le Figaro*. However, this newspaper is distinguished by the highest frequency of reported speech (68.6%) as opposed to simple mentions, whether they appear directly (70.2% of reported speech) or indirectly (65.3% of reported speech). The regional daily *L'Express de Madagascar* also has high proportions of reported speech (66.8%, as compared to simple mention), but this is mainly the case for the actors mentioned directly (74% of reported speech) and does not concern as much those mentioned indirectly (40% of reported speech). The local daily *Mayotte la 1ère* has the lowest rates of reported speech, with 33.6% on average, followed by the regional daily *Journal de l'Île de la Réunion* (42.7%) and the national daily *Le Monde* (45.5%). Again, the proportion of reported speech is invariably lower among actors indirectly mentioned by a third party than for actors appearing directly.

**TABLE 3: Reported speech vs simple mention in each of the 6 newspapers selected. In a database of 343 press articles published from 10/05/2018 to 10/05/2021, we identified actors that played a part in the information chain regarding the seismo-volcanic activity off the coast of Mayotte. Indirect mention refers to when an actor is introduced in the press discourse by a third party as opposed to direct mention. A distinction is drawn between actors whose speech or opinion is reported, when anything presented as their word or opinion is reported, even distorted, and actors that are simply mentioned.**

| | Number of news items | Number of actors mentioned | Average number of actors mentioned per news item | Direct mention | | | Indirect mention | | | | Share of direct mentions in total | Share of reported speeches in total |
|---|---|---|---|---|---|---|---|---|---|---|---|---|
| | | | | Reported speech | Simple mention | Total | Reported speech | Simple mention | Total | | | |
| Journal de | 190 | 2541 | 13 | 1112 | 825 | 1937 | 185 (30.6%) | 419 | 604 | | 76.2% | 51.0% |



| | | | | | | | | | | |
|---|---|---|---|---|---|---|---|---|---|---|
| Mayotte | | | | (57.4%) | (42.6%) | | | (69.4%) | | |
| Mayotte la 1ère | 82 | 999 | 12 | 292 (36.4%) | 511 (63.6%) | 803 | 44 (22.4%) | 152 (77.6%) | 196 | 80.4% | 33.6% |
| L'Express de Madagascar | 25 | 259 | 10 | 151 (74.0%) | 53 26.0%) | 204 | 22 (40.0%) | 33 (60.0%) | 55 | 78.8% | 66.8% |
| Journal de l'Île de la Réunion | 21 | 422 | 20 | 160 (44.9%) | 196 (55.1%) | 356 | 20 (30.3%) | 46 (69.7%) | 66 | 84.4% | 42.7% |
| Le Figaro | 15 | 296 | 20 | 139 (70.2%) | 59 (29.8%) | 198 | 64 (65.3%) | 34 (34.7%) | 98 | 66.9% | 68.6% |
| Le Monde | 10 | 310 | 31 | 132 (51.8%) | 123 (48.2%) | 255 | 9 (16.4%) | 46 (83.6%) | 55 | 82.3% | 45.5% |


Beyond these structural characteristics, the media are distinguished by the place given to different categories of actors
(Fig. 2), revealing specialisations in the event narration. If the actors linked to scientific research and monitoring (groups,
publications and institutions) are the most present in all the newspapers, this proportion varies from one media to another:
they are prevalent within the regional daily *Journal de l'Île de la Réunion*, and, to a lesser extent, within *Le Monde*, while
they only represents less than a quarter of the actors mentioned by the national daily *Le Figaro*. This trend is reinforced if
we add up the scientific and monitoring actors whose names are explicitly mentioned. The second category of actors most
represented in the different media is that of risk and crisis management actors. Again, this proportion varies from one
media to another (the highest rates are observed in the regional daily *L'Express de Madagascar* and in the national daily
*Le Figaro*). Interestingly, the populations at risk in Mayotte do not exceed the quarter (or even the tenth in certain
newspapers such as the local daily *Journal de Mayotte*, the regional daily *Journal de l'Île de la Réunion* or *Le Monde*) of
the shares of actors mentioned in the media. The relative presence of other groups of actors is more variable from one
media to another: for example, citations of social networks are more common in *Le Figaro* than in other newspapers, and
we observe a smaller frequency of actors from the French political institutions category in local newspapers than in
national and regional newspapers (with the clear exception of *Le Figaro*).





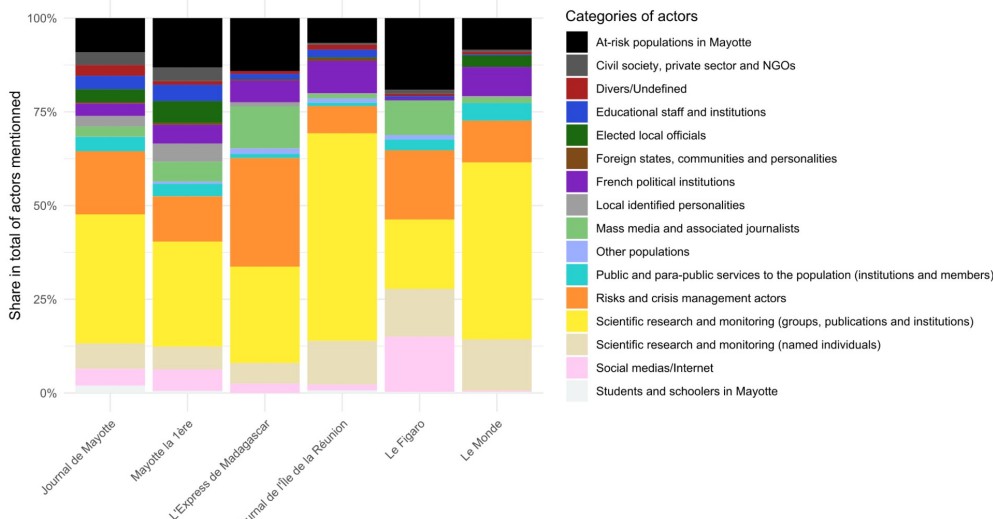


**Figure 2: Variations in relative share of crisis actor categories in all publications by media. Categories were inspired from the organisation of risks and crisis management in France (see Fearnley et al., 2018 and Section 2 in Part 2).**


**4.2 Direct speech opportunity vs framing of speech / reported speech**

TABLE 4 reports the volume and share of direct or indirect mentions, and of reported speech or simple mentions, by
category of actors in the entire corpus. Each category stands out as being mainly named in the news without an
intermediary (i.e. through a direct mention), but for certain groups, their mentions in an article consecutive to a third party
(i.e. indirect mention) are more common than for others. This is particularly the case for Mayotte's populations, which, in
more than 4 cases out of 10, are indirectly mentioned in the articles. The share of direct mentions is also relatively lower
(to a lesser extent) for public and para-public services to the population (institutions and members) and for the students
and schoolers in Mayotte (more than 3 cases out of 10) than for other groups. Named scientific and monitoring actors and
local identified personalities groups are the ones that are the most frequently directly mentioned in the corpus.
There are large variations between categories of actors in terms of relative importance of reported speech compared to
simple mentions. The analysis shows that scientific, media and institutional actors benefit from more frequent reported
speech than lay people. For instance, the total proportion of reported speech for students and schoolers in Mayotte is only
10.8% and 32.4% for the at-risk populations in Mayotte, while it reaches 62.9% for local personalities, 77.2% for
scientific research and monitoring named individuals, 78.8 % for mass media and associated journalists and 83.3% for the
social media/Internet. Among the categories identified, only named scientific actors and foreign states, communities and
personalities have a higher share of reported speech when mentioned indirectly by a third party than by a direct mention
in the article, which seems to illustrate the importance of these external points of view on the perception of the crisis by
the various local actors.






**TABLE 4: Key figures on the mentions of categories of actors in all the news items of the corpus. In a database of 343 press articles published from 10/05/2018 to 10/05/2021, we identified categories of actors that could play a part in the information chain regarding the seismo-volcanic activity off the coast of Mayotte. Indirect mention refers to when an actor is introduced in the press discourse by a third party as opposed to direct mention. A distinction is drawn between actors whose speech or opinion is reported (anything presented as their word or opinion, even distorted) and actors that are simply mentioned.**

| | Number of mentions | Direct mention | | | Indirect mention | | | Share of direct mentions in total | Share of reported speeches in total |
|---|---|---|---|---|---|---|---|---|---|
| | | Reported speech | Simple mention | Total | Reported speech | Simple mention | Total | | |
| Scientific research and monitoring (groups, publications and institutions) | **1762** | 633 (43.7%) | 815 (56.3%) | **1448** | 100 (31.8%) | 214 (68.2%) | **314** | **82.2%** | **41.6%** |
| Risks and crisis management actors | **683** | 334 (63.9%) | 189 (36.1%) | **523** | 58 (36.3%) | 102 (63.7%) | **160** | **76.6%** | **57.4%** |
| At-risk populations in Mayotte | **549** | 122 (37.7%) | 202 (62.3%) | **324** | 56 (24.9%) | 169 (75.1%) | **225** | **59.0%** | **32.4%** |
| Scientific research and monitoring (named individuals) | **355** | 237 (76.0%) | 75 (24%) | **312** | 37 (86.0%) | 6 (14.0%) | **43** | **87.9%** | **77.2%** |
| French political institutions | **208** | 85 (53.1%) | 75 (46.9%) | **160** | 11 (22.9%) | 37 (77.1%) | **48** | **76.9%** | **46.2%** |
| Social media/Internet | **186** | 134 (90.5%) | 14 (9.5%) | **148** | 21 (55.3%) | 17 (44.7%) | **38** | **79.6%** | **83.3%** |
| Mass media and associated journalists | **179** | 120 (81.6%) | 27 (18.4%) | **147** | 21 (65.6%) | 11 (34.4%) | **32** | **82.1%** | **78.8%** |
| Public and para-public services to the population (institutions and members) | **148** | 31 (34.1%) | 60 (65.9%) | **91** | 7 (12.3%) | 50 (87.7%) | **57** | **61.5%** | **25.7%** |
| Educational staff and institutions | **135** | 29 (31.5%) | 63 (68.5%) | **92** | 11 (25.6%) | 32 (74.4%) | **43** | **68.1%** | **29.6%** |
| Civil society, private sector and NGOs | **129** | 39 (39.8%) | 59 (60.2%) | **98** | 4 (12.9%) | 27 (87.1%) | **31** | **76.0%** | **33.3%** |
| Local identified personalities | **116** | 66 (66.0%) | 34 (34.0%) | **100** | 7 (43.7%) | 9 (56.3%) | **16** | **86.2%** | **62.9%** |
| Elected local officials | **111** | 47 | 38 | **85** | 5 (19.2%) | 21 | **26** | **76.6%** | **46.8%** |



| | Number of mentions | Direct mention | | | Indirect mention | | | Share of direct mentions in total | Share of reported speeches in total |
|---|---|---|---|---|---|---|---|---|---|
| | | **Reported speech** | **Simple mention** | **Total** | **Reported speech** | **Simple mention** | **Total** | | |
| | | (55.3%) | (44.7%) | | | (80.8%) | | | |
| Students and schoolers in Mayotte | **65** | 6 (13.3%) | 39 (86.7%) | **45** | 1 (5.0%) | 19 (95.0%) | **20** | **69.2%** | **10.8%** |
| Divers/Unidentified | **64** | 56 (98.2%) | 1 (1.8%) | **57** | 1 (14.3%) | 6 (85.7%) | **7** | **89.1%** | **89.1%** |
| Other populations | **20** | 7 (63.6%) | 4 (36.4%) | **11** | 1 (11.1%) | 8 (88.9%) | **9** | **55.0%** | **40.0%** |
| Foreign states, communities and personalities | **20** | 5 (33.3%) | 10 (66.7%) | **15** | 3 (60.0%) | 2 (40.0%) | **5** | **75.0%** | **40.0%** |


**4.3 Position of actors in the citation network**
The cross-analysis of the identification frequencies of actors as source or recipient of a quotation in the corpus (Figure 3)
indicates that the role played by the prefecture of Mayotte and its main representatives in communicating about the event
is central, since their appearances in the media largely lead to the mention of other actors. Many mentions of actors in the
event also come from the Twitter network, which appears to be an essential primary source in the media story. A few
local personalities also emerge as central nodes of the network, and make it possible to relay information concerning a
large number of actors. This is the case of: i) Saïd Saïd Hachim, a Mahorese geographer working at the Departmental
Council of Mayotte also achieving a PhD in geography at Paul Valéry Montpellier 3 University in mainland France; ii)
Lieutenant-Colonel Philippe Blanc, a member of the Directorate-General for Civil Protection and Crisis Management
(part of the Bureau for Exercise Planning, Feedback and Coordination of the Beauvau Crisis Centre in the Directorate-
General for Civil Protection and Crisis Management.) who exercices at a national level and was sent in Mayotte in June
2018 as a member of an interministerial delegation for civil protection in the context of the seismic crisis; iii) Eric
Humler, scientific director of REVOSIMA (volcanological and seismological monitoring network in Mayotte) from 2019
to 2022 and in charge of the coordination of the TelluS-*Mayotte* mission, and iv) Frédéric Tronel, the regional director of
BRGM (French geological survey BRGM) in Mayotte from 2017 to 2020. The UDAF (Departmental Union of Family
Associations) interestingly emerges as a key player in the chains of citation of the actors within the corpus. This can be
explained by the meeting they organised on the 5[th] of June 2018 to relay people's experiences and promote dialogue
between local actors (among them, state institutions, public and para-public services to the population, local elected
representatives, ect) regarding the measures to be taken at the start of the seismic crisis (*Le Monde*, 14/06/2018 and
*Journal de Mayotte*, 01/06/2018) and which was relayed by both local and national press. On the contrary, the Mahorese
population and, to a lesser extent, the schoolchildren, highly cited but not often at the origin of the citation of a third
actor, indicate a relatively passive position within the citation network extracted from the corpus. This is also the case for

28                                        14



the French geological survey BRGM, the REVOSIMA (Volcanological and seismological monitoring network in
Mayotte) and the scientific community (in general), which are regularly used as a source of information by third party
actors whose words are reported (directly or indirectly) in the articles.

Figure 4 shows an exponential relationship between the normalised values of the betweenness indices and the ranks,
indicating a hierarchical structure of the network through the concentration of citation interactions by a small proportion
of actors. The Mahorese population emerges as the key connection element of the network of actors constructed from
citation links in the media, a role also highly played by the local expert Saïd Saïd Hachim. The central role of the prefect
and the prefecture, representing the French State in the department, is also depicted. The scientific community as a whole,
the French geological survey (BRGM) and the REVOSIMA also appear to be essential elements in the structuring of the
network, so is the online social media Twitter allowing media visibility for individuals and institutions, and promoting
citation chains via re-tweets and identifications. Expectedly, individuals with many connections also have high degrees of
betweenness: Philippe Blanc, Eric Humler, Frédéric Tronel mentioned above as well as Nathalie Feuillet, observatory
physicist at IPGP (Institut de physique du globe in Paris) and mission leader of the first oceanographic campaigns
(MayObs 1 and 2), who, despite lower in-degree and out-degree values, contributes to concentrating a relatively large
number of shortest paths in the network.

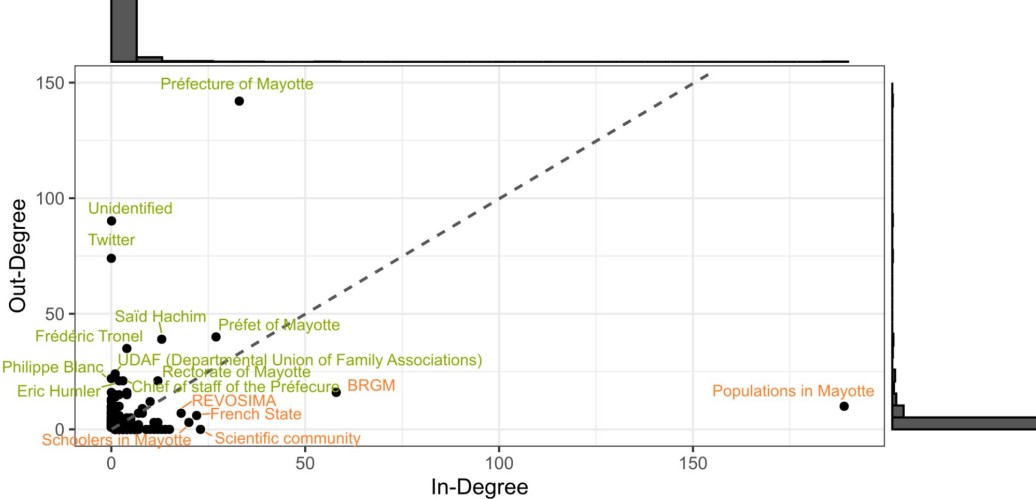


**Figure 3: In- and out-degree distributions per category of actors in all the news items of the corpus.**
**Scatterplot of the number of times an actor is mentioned at the start of a quoting chain (out-degree) over the number of times**
**he or she is mentioned as a recipient of a quoting chain (in-degree). The actors who are most often the source of the quote (and**
**who are more often the source than the one cited) are presented in green, while the actors who are most often the subject of the**
**quote (and who are more often the one cited than the source of a quote) are presented in orange.**






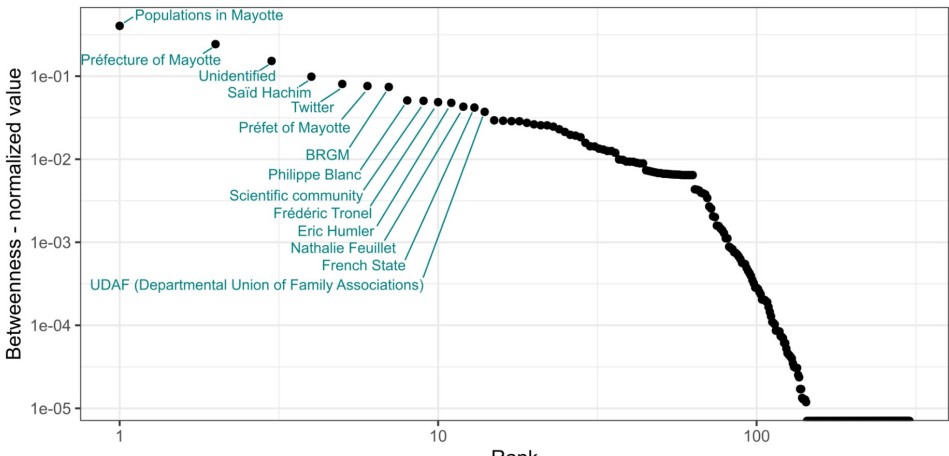

**Figure 4: Rank size distribution of betweenness centrality values. The plot shows the relationship between betweenness**
**normalized values of actors and ranks (in logarithmic scales).**

**4.4 Actor network structure**

The 126 articles published during the first months of the crisis (period 1, from 05/10/18 to 07/26/18) reflect a network of
highly interconnected actors structured around actors polarising a certain number of citations (Figure 5): these are the
Mayotte population as a whole, the prefecture of Mayotte and the prefect, the Departmental union of family affairs, actors
of the educational system (rectorate, teachers), but also mayors (association of mayors, mayor of Mamoudzou-the capital
city, mayor of Chirongui, or other municipalities) and the social network Twitter. The municipality of Chirongui stands
out here, probably because its hosting of the Groupe d'Intervention Macrosismique served as an entry point for the
presentation of the group in the local press, what's more, the mayor at the time remained in office from 2008 to 2020 and
her team seems to have been particularly active and well integrated into the local community. The citation network is
extensive, and all categories of actors are present with the exception of local personalities (although many groups made
up of members from civil society, associations and businesses appear in the network). Interestingly, if the citation chains
from the entire corpus show paths between actors belonging to the same category, the network itself presents a certain
heterogeneity. The prefecture of Mayotte relayed information from local and other national risks and crisis management
actors, but also from scientific research and monitoring institutions, groups and publication, and was cited by and
communicated through a variety of mass media and social media. The articles in the corpus also make it possible to link
the prefecture of Mayotte to the Mahorese population, as well as to various public and parapublic services such as
hospitals. Thus the prefecture emerges as the central actor in the management of this first period of the seismo-volcanic
activity, which is in accordance with the missions of security of people and property and representation of the State which
are conferred. It is interesting to note that the prefect's interventions and mentions in the press do not link him to the same
actors as the institution he represents. His communications aimed at the civilian population and representatives of civil
society and associations, stakeholders in the educational world, and mayors. Actors from the delegation of specialists in
civil security and natural risks like Mendy Bengoubou appear as intermediate nodes between the prefecture and the





prefect, with whom they share a large number of co-cited actors. Conversely, in these first months of the crisis, the
citation networks between scientific actors appear fragmented and local elected officials relatively peripheral.

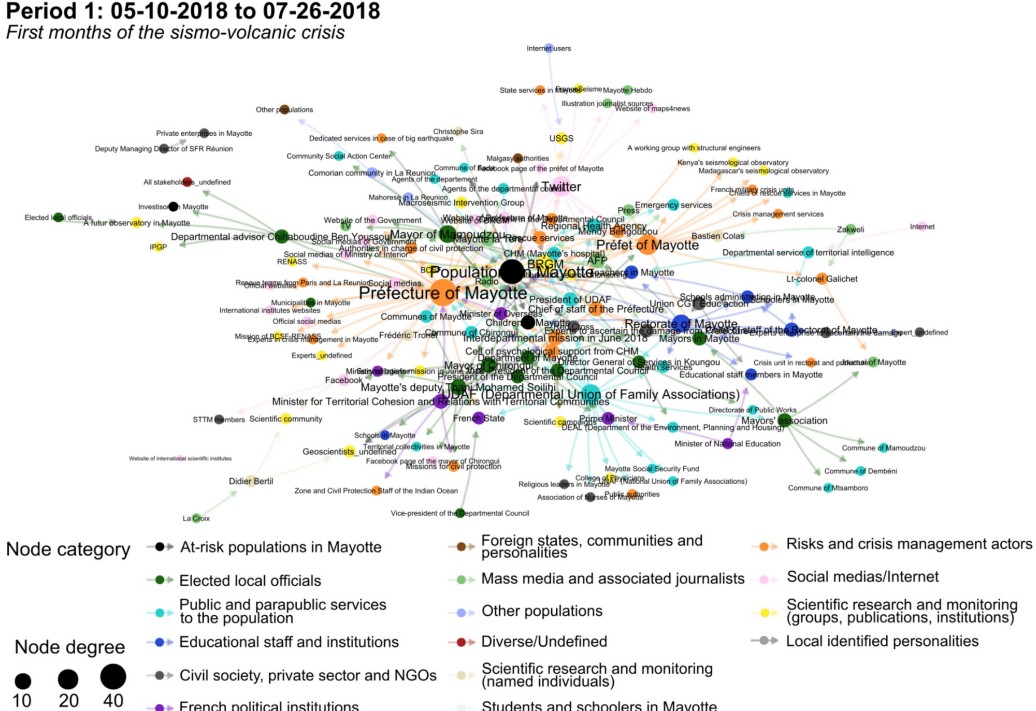

Figure 5: The citations network of actors in the press during the first months of the sismo-volcanic crisis

The network of citations (Figure 6) for the second period (selection of 29 articles mentioning and discussing the volcanic
hypothesis and the first subsidence data between October 05th, 2018 and July, 26th 2018). Just as for the first period of
the crisis, the population (cited by numerous actors, but never at the origin of a single quote) finds itself at the centre of
the network. The prefecture and the prefect (and to a lesser extent the chief of staff of the prefect) still emerge as central
nodes of the network, as the origin of numerous citations. Only the prefecture, as an institution, receives a significant
number of quotes in return, from various categories of actors. Once again, the institution is separated from its two key
figures (prefect and chief of staff) within the citation network, with the exception of common citations to the population
and the BRGM. The BRGM is also the destination of numerous citations from scientific actors, which it allows to
partially aggregate within the network. Two scientific personalities stand out for the plurality of co-citation links they
create: Laure Fallou and Frédéric Tronel. The first is a sociologist research officer at EMSC (Euro-Mediterranean
Seismological Centre) who wrote an academic paper calling attention to the emergence of a mistrusting atmosphere and
circulation of misinformation due to a lack of scientific information linked with the scarcity of seismic data. Fallou et al.
(2020) was published in 2020, but it has also been the subject of a public communication at the General Assembly of the
European Geosciences Union in April 2019. The second was the regional director of BRGM in Mayotte between 2017 to



2020. In contrast, Nathalie Feuillet as an observatory physicist, the University of La Rochelle in mainland France and
Tellus Mayotte oceanographic mission from IPGP are isolated. Christophe Sira, a macroseismic surveyor member of the
Macroseismic Intervention Group, and Laurent Michon, a research professor at University on La Réunion island, are also
separated from other scientific actors. Likewise, Bastien Colas and Mendy Bengoubou, 2 out of the 3 members of the
delegation of specialists in civil security and natural risks dispatched by two ministries (Ministry of Ecology and Ministry
of the Interior) to assess the seismo-volcanic activity on site in June 2018, are cited separately (and are separated from
mentions of the interdepartmental mission they belonged to), while the third expert, Lieutenant-Colonel Olivier Galichet
is not mentioned in the selected articles. Once again, the actors in the medico-social world, at the origin of citations to
various actors, have intermediate positions in the network, while the actors in the educational world find themselves more
isolated than during the previous period. Another important distinctive element is the introduction of a first identified
local personality, Saïd Saïd Hachim, in the network, which only cites the prefecture in this sub-corpus of the article as he
relayed and detailed a note from the prefecture mentioning the volcanic origin hypothesis derived from the latest GPS
data. Local fishermen also appear in the network of actors via citations from IFREMER and the chief of staff of the
prefecture, following the discovery of deep-sea dead fishes about 50 km eastward from coast.

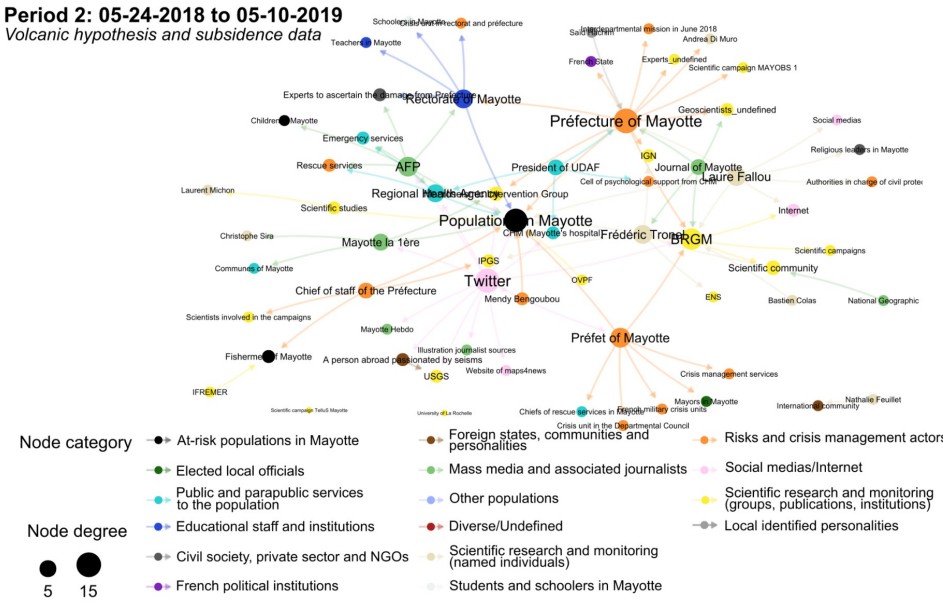

**Figure 6: The citations network of actors in the press after the emergence of the volcanic origin hypothesis and the first**
**subsidence data.**

The third period studied corresponds to the discovery of the underwater volcano and concerns 36 articles published from
May 16th, 2019 to August, 30th 2019. Figure 7 depicts its actor co-citation network. As compared to the previous
periods, the catalytic role of the population is diminishing, while the position of local actors acting either as relays for
residents' voices such as the Facebook group Signalement Tremblement de Terre Mayotte (S.T.T.M, for "earthquake



37

reports in Mayotte" which was created after the first earthquakes of 2018) or as relays for scientific voices such as Saïd
Saïd Hachim is strengthened. Philippe Blanc, as a member of a Civil Security mission on volcano-related risks, also
appears as an important source of citations in the network, mainly for other risks and crisis management actors, without
benefiting from any quote from a third party in return. This is also the case of Jean-Michel Audibert, who was part of the
same mission for civil security, without the two actors sharing any other common citation than that of the Mahorese
population. The various ministries involved in crisis management and the French state are only indirectly connected to
each other, reflecting segmented cooperation networks even within the framework of inter-ministerial actions. The
Ministry of Overseas Territories introduces numerous actors into the citation network, including REVOSIMA (Mayotte's
volcanological and seismological monitoring network) which was just created on the 18th of June 2019 and a hypothetical
future observatory in Mayotte which is still in the planning stage at the time of writing. The network of scientific actors is
more structured and includes more numerous, diverse and international actors than in the previous periods. Indeed, the
recording of the waves of a VLP earthquake all over the world on 11/11/2018 has attracted the attention of international
institutions and media in addition to that of scientific and political authorities at national level (Hossein and Sadaomi,
2021). It is however interesting to note that the scientific actors named by Eric Humler or via the Twitter network differ
from those named by Saïd Saïd Hachim, which mainly relays the names of scientists with whom he had published an
atlas of natural risks and vulnerabilities of Mayotte in 2014 and discusses MayObs' campaigns. Several social networks,
the Facebook group S.T.T.M and two local newspapers appear as important nodes of the citation network, mainly citing
other media, the Mahorese population, scientists (despite for the STTM Facebook group), public institutions, and the
actors of the MayObs oceanographic campaigns carried out from the Ifremer ship Marion Dufresne. Interestingly, the
member of the MayObs oceanographic campaigns are not cited by the same actors, reflecting a sequencing of the actors
mentioned in the media as the oceanographic campaigns progress (the Prefecture, the Préfet and Twitter for the 1st
campaign, the Ministry of Overseas and Saïd Saïd Hachim for the 4th, while interministerial communiqués do not
distinguish between the different missions).



39

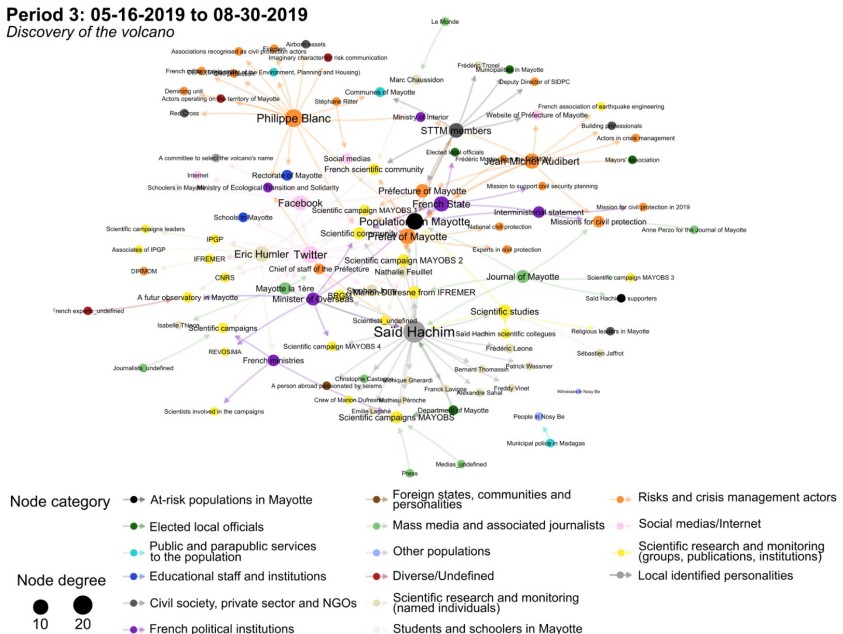


**Figure 7 : Citations network of actors in the press after the discovery of the volcano.**

Period 4 focuses on a particular moment intersecting period 3: the organisation of a conference for local elected officials
and the press organised by the prefecture and led by scientists who participated in the discovery of the volcano. It
includes 6 articles published between July 31, 2019 and August 9, 2019. As shown in the network of actors represented in
Figure 8, Nathalie Feuillet plays the role of intermediary between, on the one hand, the part of the network structured
around Saïd Saïd Hachim and that structured around the STTM Facebook group with which she is indirectly linked via
his quote of the prefecture. Without this, the network of actors appears fragmented, even between actors at the heart of
the event: the scientists, the representation of the French State via the prefecture, the prefect and his chief of staff, but also
via the Interministerial Defense and Civil Protection Service (SIDPC), just like the Ministry of Overseas Territories and
the Ministry of Ecological and Inclusive Transition. Frédéric Tronel and Isabelle Thinon are the only scientists named by
a third party, the STTM Facebook group for the first and the local news broadcast Mayotte la 1ère for the second. Added
to this low presence of named scientific actors is the absence of a mention of specific scientific institutions, in favour of
abstract mentions of the community.



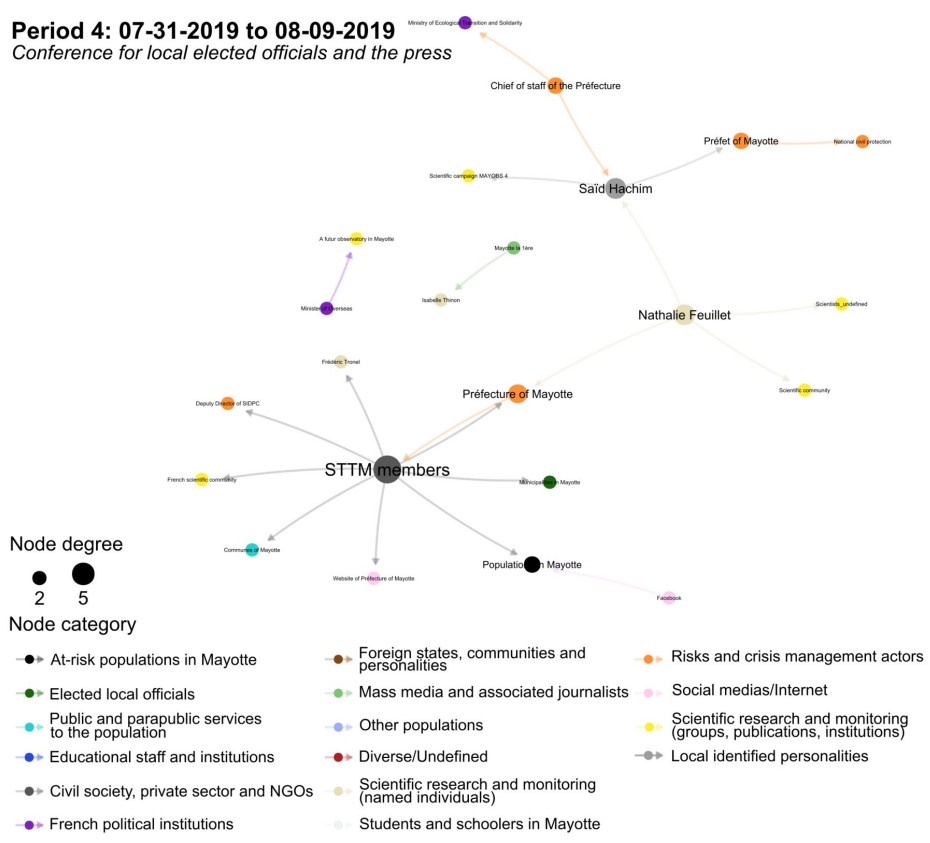

**Figure 8: Citations network of actors in the press following the conference for local elected officials and the press organised by the prefecture.**

Period 5 focuses on a second particular moment intersecting period 3: the visit of the Overseas minister in Mayotte. It includes 3 articles published between August 27, 2019 and August 30, 2019. As depicted in Figure 9, the number of cited actors is low. The Minister of Overseas occupies a central place in the network since it mentions 5 other actors: the scientific actors in charge of studying the phenomenon (REVOSIMA and the observation campaign underwater MayObs 4), and the scientific community more generally, as well as the local expert Saïd Saïd Hachim and the Mahorese population, also cited by the prefect. We also note the isolated mention of the French State by Frédéric Mortier (from the interministerial delegation for major overseas risks - DIRMOM) and the unrelated mention of local elected officials.



43

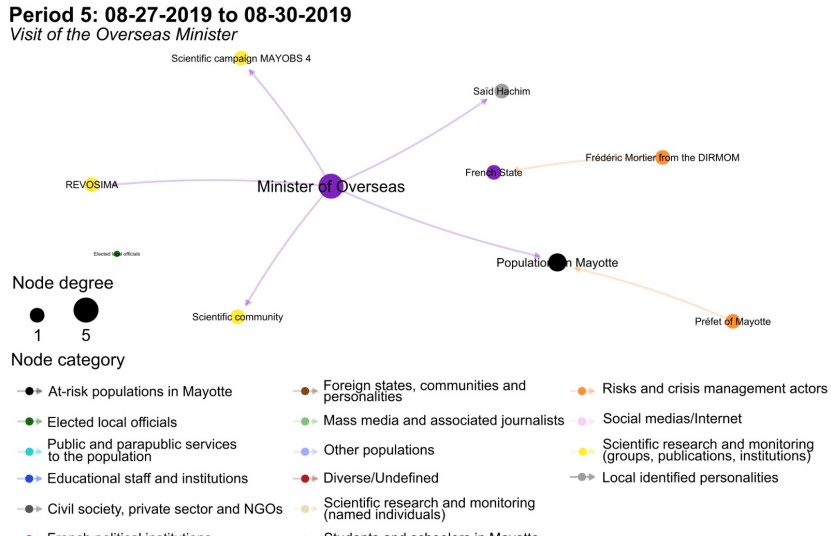

530

**Figure 9: Citations network of actors in the press following the visit of the Overseas Minister in Mayotte.**

532

8 articles in the corpus cover the period from September 4, 2019 to October 26, 2019, during which a scientific conference is held at IPGP headquarters in Paris and immediately followed by a public conference operated by scientists and ministerial officials. Interestingly, REVOSIMA (once again solely cited by the Ministry of Overseas Territories) is isolated from the rest of the citation network (Figure 10), even though it brings together the actors in charge of volcanological and seismological monitoring of Mayotte. It is nevertheless the only scientific institute explicitly named, the citation network giving a more central place to individual scientists than to institutions. The network of actors is more fragmented than previously, and its subparts are respectively organised around Nathalie Feuillet (who received citations from various media), Eric Humler (quoting populations from other French overseas departments), the chief of staff of the Préfecture (citing the communes of Mayotte and territorial authorities in general), and the journal *Sciences et vie* (citing Nathalie Feuillet and encompassing the scientific community and French research institutes). Several isolated citations point again to the Mahorese populations (from scientists Virginie Duvat and Frédéric Tronel, but also from the Préfecture of Mayotte and from a local elected official from the capital city Mamoudzou). Once again, the prefecture of Mayotte is separated from the prefect and his chief of staff within the citation network. For the first time, the Prime Minister is cited by an actor, namely the deputy in charge of development in Mamoudzou, regarding a letter the latter sent to the Prime Minister to point out the effects of subsidence on urban development.

44





45

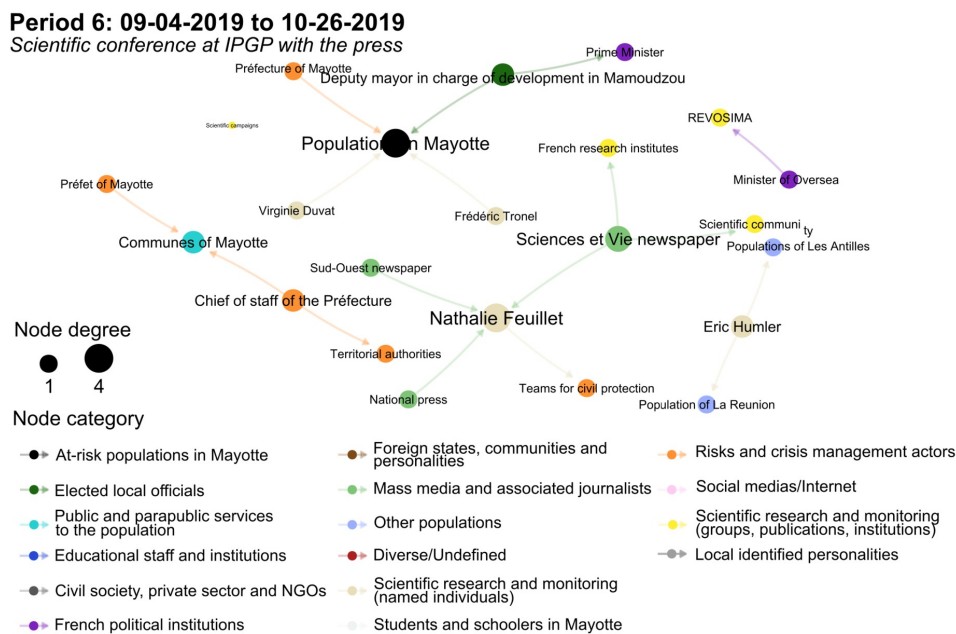

**Figure 10: Citations network of actors in the press following the scientific conference at IPGP followed by a public conference by scientists and ministerial officials and attended by journalists.**

Period 7, which extends from May 4, 2020 to September 28, 2020 and covers the missions MayObs 13-1 and MayObs 13-2, includes 7 articles from the corpus. Its citation network is shown in Figure 12. For the first time, REVOSIMA appears as a central node of the network, although the number of nodes and links is moderated by the small number of articles. The REVOSIMA is cited by local expert Saïd Saïd Hachim and by the Prime Minister, but also by the BRGM, one of its membering institutions. The Prime Minister also cites the IPGP separately from REVOSIMA, even though the institute is in charge of it. The network of scientific actors appears generally fragmented during these two missions. It should also be noted that the documentary "Birth of a volcano" produced by *Crestar Productions* and *L'éolienne* and broadcasted on *Mayotte la 1ère* is not relayed by any actor other than the channel.

The last period extends from October 28, 2020 to November 3, 2020 and deals with two events recounted in 6 articles: the "Volcano week" and the first siren alerts in the municipality of Dembéni. The network of actors (Figure 12) is structured around two parts. The first is organised around Mahorese schoolchildren, and includes quotes from the chief of staff of the Rectorat of Mayotte, from the chief of staff of the Préfecture and from the Préfecture itself. The teachers in Mayotte are part of the citation chain, as they are also quoted by the chief of staff of the Préfecture. Interestingly, schools in Mayotte (mentioned as actors) belong to the second citation subnetwork, of which they form the periphery thanks to a citation by Charlotte Mucig, herself cited by the TV and radio show Zakweli broadcasted by Mayotte la 1ère. The show Zakweli also makes the link with Eric Humler and Frédéric Mortier, both citing the Mahorese population. The latter is also cited by Mahorese associations not identified in the articles, as well as by scientists visiting Mahoran schools during





47

the volcano week, who will also mention the MayObs missions on the ship Marion Dufresne more or less explicitly.
Finally, REVOSIMA appears fully isolated from the other actors.

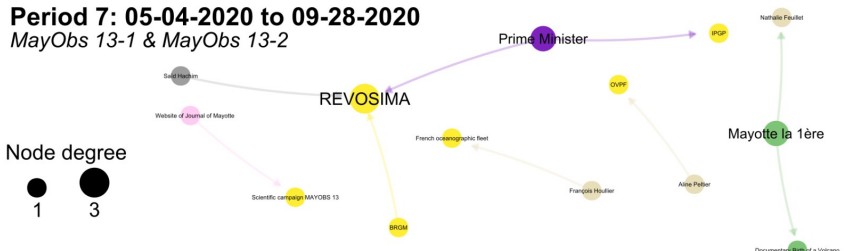

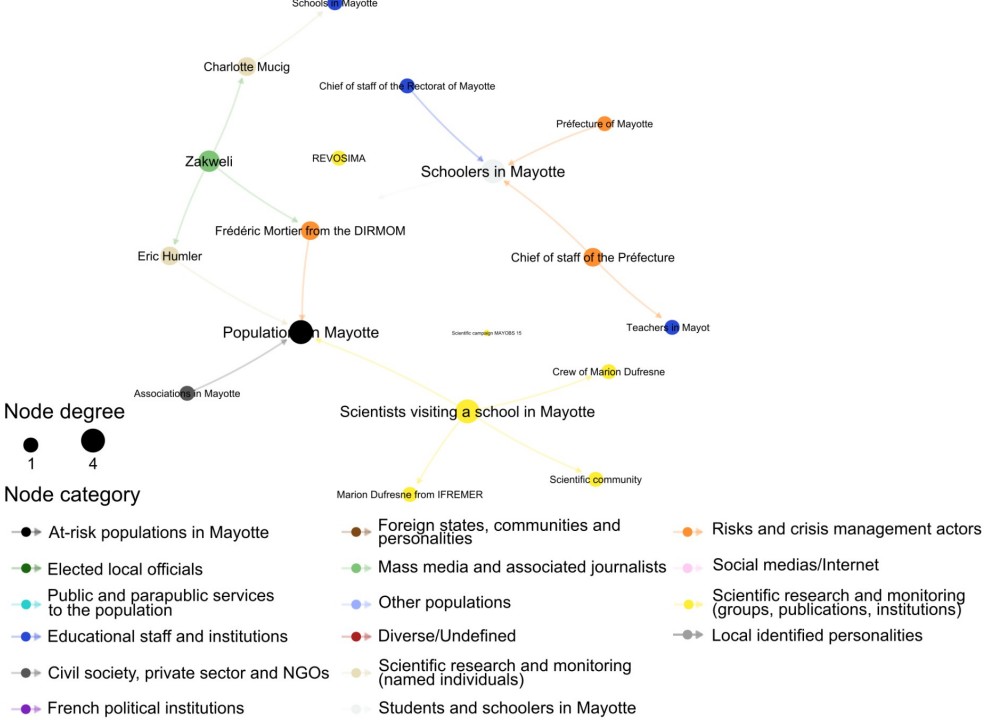


**Figure 11: Citation networks of actors during two press-covered events : In Period 7, the focus lies on the organisation and**
**execution of two complementary campaigns at sea, MayObs 13-1 and MayObs 13-2. In Period 8, attention shifts to two**
**communication initiatives from the Mayotte prefecture, namely the "Volcano week"  a serie of conferences and activities**
**related to the discovery of a new volcano designed for schoolers, and the installation of a first siren alert on the territory,**
**widely relayed by the media.**



49

**5. Discussion**

Mass media play a key role in risk and crisis communication, serving as the main information source for millions of
people regarding natural, political and social events (e.g. Gamson and Modigliani, 1989; Allan et al., 2000; Dixon et al.,
2008 ; Aday, 2010). They influence people's perceptions of various actors involved in understanding, monitoring hazards,
and managing their effects, as well as their performance during events (Harris et al., 2012). Despite several limitations in
our study, such as the focus on newspaper representations rather than those among populations, and the use of articles
from six non-specialist French-language newspapers, it provides a comprehensive insight into media narratives during the
seismic-volcanic "crisis" in Mayotte from spring 2018 to spring 2021. The inclusion of additional local, regional, and
national daily papers in different languages and other media forms could enhance the depth of the dataset. However, the
chosen corpus, primarily non-specialist daily press (including *Mayotte la 1ère*, which displays a TV, radio, a written
website, and produces content in both French and Shimaore thus widely followed by inhabitants in Mayotte), remains a
strong candidate for studying representations conveyed by the media as a whole (Cagé et al., 2017), especially
considering the dynamics and complementarity of quantitative and qualitative analyses facilitated by written press data.

When examining the results, it is crucial to acknowledge the influence of journalistic writing. Newspapers, acting as
mediators in the broadest sense, have their own practices and priorities in collecting and disseminating information. Daily
newspaper journalists, often generalists with diverse profiles and working methods (Ruellan, 1992), are commonly bound
by the shared constraint of tight deadlines. This limitation may affect their ability to access multiple sources or delve
deeply into the context. Consequently, they tend to prioritise information that is deemed reliable, easily accessible, and of
interest to their readership (van Belle, 2015). Geographic proximity to events influences coverage (e.g. Cavacas et al.,
2016), with local journalists having easier access to the field and local actors. This is evident in the composition of the
selected articles, with a predominant 78% (and 54% in *Journal de Mayotte* alone) being published in local newspapers.
These local articles typically take the form of concise press releases offering updates on the latest developments.
Contrastingly, national newspapers, faced with the challenge of engaging a readership distant from about 8, 700 km and
not directly impacted by events, tend to favour less frequent but more extensive publications providing summaries or in-
depth analyses (see section 4.1: an average of 20-31 actors are mentioned per article in national newspapers, compared to
around ten in local newspapers). This also influences the selection of quoted sources, with local journalists relying more
on local actors and national journalists adopting alternative strategies like using social networks (here Twitter and
Facebook for the national dailies *Le Figaro* and *Le Monde*) or relying on news agencies like AFP (France-press Agency)
or their local counterparts (Lecheler and Kruikemeier, 2016). This is illustrated here by *Le Figaro* and *Le Monde* which
can present a comprehensive view of the involved actors, albeit with less ease in reporting their statements. Given their
limited time constraints, journalists typically favour sources they consider legitimate and relatively easy to access. The
choice of sources may vary based on editorial stance (Wang et al., 1992; Shoemaker and Reese, 1996). For example, in
this context, local newspapers, which are seemingly inclined to emphasise the local context in comparison to national and
regional newspapers exhibit a lower rate of reported speech from French political institutions. As observed in our findings
and reinforced by Ploughman (1997), journalists tend to have more accessible and regular contact with certain types of
sources, including public institutions, officials, or high-profile personalities, which are more echoed in the news.

Having acknowledged the influence of journalistic writing, we show that using this method to scrutinise press coverage
also allows the identification of actor groups typically present in a crisis context related to a natural phenomenon (e.g.

50



51

Fearnley et al., 2018; Trias et al., 2019, using network analysis on disaster risk reduction ecosystem in the Asia Pacific
region; Gonzalez, 2022, using assemblage theory and mapping relationships following the 1985 San Antonio
Earthquake). First of all, the main trio in crisis management emerges as the most mentioned and quoted actors in this
network: scientists overseeing monitoring, authorities responsible for civil protection, and at-risk populations (Fearnley et
al., 2018; Devès et al., 2023). Other categories, such as mass media, social networks, civil society, public and para-public
services, and even humanitarian aid associations (e.g. the Red Cross), are also well-represented. However, the latter are
less prevalent than in other crises, likely due to relatively minimal material and human damage (only three lightly injured
and cracked buildings in Mayotte). Notably, international actors, including personalities, newspapers, communities and
states, are also present, which is highlighted in previous studies as indicative of a growing interconnection between actors
in disaster risk reduction context on an international scale (e.g. Trias et al., 2019). In the case of this very local, small-
scale crisis, the mere presence of international reactions becomes an event worth reporting by others. The actors involved
in crisis management, organised by the ORSEC (Civil Security Response organisation) in France, are also represented.
As a reminder, according to ORSEC framework, mayors coordinate emergency services and public facility management
(hospitals, schools, etc.) if the event is local. If damages extend department-wide and surpass the capacity of town halls,
the prefecture assumes control. Ultimately, the crisis is managed by the regional headquarters (EMZCOI for Mayotte),
then by the French government if the lower levels are overwhelmed. However, here despite the limited physical impact of
this crisis, it is mainly the prefecture and national civil protection services (under the responsibility of ministries) that are
mentioned and whose speech is reported. Local elected representatives, including mayors, are surprisingly less present
than other actors who theoretically play a lesser role (such as civil society, local personalities, and public and semi-public
services, including education staff), while the regional headquarters is rarely mentioned. These asymmetries in
representation partly reflect Mayotte's unique situation as a French overseas department, with part of its administration,
including the regional headquarters, based more than 1,400 km away in Reunion Island. This region, while distant from
the mainland, poses specific challenges requiring a substantial response from the national authorities responsible for crisis
management (Cottereau, 2021; Roinsard, 2022; Duchesne, 2023). Another specific feature also observed in several
overseas departments is a lesser degree of cooperation between local elected representatives and national representatives
located in these departments (Lemercier et al., 2014; Gillet et al., 2023). Additionally, our results highlight the emergence
of different actors or groups of actors within the population, a facet not treated in the press as homogenous, as seen in
official/legal texts. Particularly, several local personalities are identifiable, as observed in other cases (e.g. Devès et al.,
2019). Overall, we obtain an overview of the diverse actors involved in this crisis management and communication,
aligning with findings from other studies using similar (e.g. Rajput et al., 2020) or different methodologies (e.g. Villodre
and Criado, 2020; Calabro et al., 2020).

Upon examining who can express themselves through media coverage, several observations emerge. Firstly, primary
sources of information (often introduced or relayed by other actors quoted in the article), do not necessarily align with
journalists' sources. This discrepancy may arise due to limited access to primary source, as elaborated earlier, either due
to geographical distance, time constraints, ect, or the preference for another source deemed as more legitimate, more
accessible or because of implicit or explicit issues of representation (see Carlson, 2009 and Grassau et al. (2021) for an
analysis of journalists' sources in an emergency situation). This opens a window into the newspaper's networks, the
hierarchy of its trust, and the perceived legitimacy of interviewed sources. As identified in a qualitative analysis (Devès,
Moirand and Le Vagueresse, 2023), scientific actors notably dominate reported speeches, both from the article authors



53

and third-party actors. Consequently, they are considered the most reliable or, at the very least, the most legitimate to
express themselves, even surpassing the authorities responsible for crisis management and civil protection. While other
studies have recognized scientists as "bridges" and "focal points" among various actors (Trias et al., 2019), the notable
overrepresentation of scientific figures and institutions is noteworthy here. This could be attributed to journalists focusing
on short-term issues like hazard descriptions, impacts, and emergency operations (Devès et al., 2019) or the complexities
arising from scientific uncertainties requiring focused attention (Valencio and Valencio, 2018). Scientific actors, being
perceived as those with the knowledge, are considered closest to understanding the phenomenon and thus are positioned
to make recommendations (Oreskes, 2019). Apart from scientific personalities and institutions, the analysis indicates that
media and institutional actors also benefit from greater media reach for their statements compared to actors from civil
society. This proximity of journalists to institutions, as highlighted in other case studies (e.g. Ploughman, 1997;
Wintterlin, 2020), and their reliance on the publications of their counterparts when direct sources are unavailable
(Coddington and Molyneux, 2023) are usual practices. On another other hand, social media sources exceed other mass
media in reported speech share, reflecting the increasing use of social media as information sources due to their detailed
coverage of current events (e.g. Lindsay, 2011; Lecheler and Kruikemeier, 2016), potentially facilitating two-way
communication between institutions and the public (Feldman et al., 2016; Kim and Hastak, 2018b). However,
Pourebrahim et al. (2019) argue that this potential is largely underused, showing that Twitter is dominated by authorities
primarily engaged in one-way communication rather than interacting with their audiences (see also Watters and Williams,
2011). A more focused analysis would be needed to explore this in the present case. Throughout the coverage,
populations in Mayotte and, to a lesser extent, schoolchildren are frequently cited, but as discussed in our qualitative
analysis, they are not often the origin of citations, indicating a relatively passive position within the citation network
extracted from this dataset. Despite this, the population is central to the network and is cited by numerous actors. In the
general imagination, its protection is the reason for the organisation of this network, but its opinion is seldom expressed,
even in local media which tend to apply pre-constructed news templates (Jemphrey and Berrington, 2000). This
perpetuates an asymmetrical and hierarchical representation favouring those perceived to hold knowledge (scientists,
institutions in charge of civil protection) at the expense of the inhabitants' perspective, who find themselves in the
position of undergoing and being protected (Valencio and Valencio, 2018; Gonzalez et al., 2022). Journalists' common
practices make it challenging for them to distance themselves from this representation (Cavaca et al., 2016). This result is
explored and assessed for this case study by Devès et al. (2023), who also demonstrate that these representations are
integrated by various actors in crisis management, particularly within at-risk populations.
That being said, it is worth underscoring that identified individuals hold a significant position in this network. Despite the
context where crisis and risk management, as well as communication, are organised and framed by established
institutions (ministries, prefectures, town halls, scientific institutions responsible for monitoring), individual sources with
clear identities tend to contribute more in reported speeches compared to institutions. Specifically, named scientists take
the lead over their institutions when we examine whose speeches these articles mostly report. Moreover, local
personalities are mainly quoted when directly mentioned, while the reverse is true when they are indirectly mentioned.
Even in the case of crisis managers, individuals such as the prefect, cabinet director or ministries envoys clearly stand out
from their respective institutions within the network (see section 4.4.). Admittedly, these personalities are often
associated with an institution. Oreskes (2019) highlights this aspect in her essay "Why trust science?", using scientists as
an example. She contends that trust in an individual is primarily conferred as a member of a professional community with



55

a shared body of knowledge. Butts et al. (2007) also found that coordinating roles, which include information flow, are
influenced by the formal institutional status or position within an organisation. Added to this is the journalistic practice of
conducting interviews, which involves collecting an institution's stance through the discourse of one of its
representatives. Individuals act as entry points to information for journalists, relaying the messages of their institutions,
colleagues, or, in all cases, a community. Hence, they play the role of hubs, or "guardian nodes" as mentioned by Flecha
et al. (2023). However, this interpretation needs qualification based on interviews with inhabitants, analysed in Devès et
al. (2023). On one hand, these interviews highlight the importance, according to Mayotte's inhabitants, of embodying
information or words with a name or a face. On the other hand, they reveal a significant distrust towards political and
even scientific institutions. In this study, we observed that featuring the director of the local BRGM branch over the
national director was prioritised, and even the prefect over representatives of the ministries, although the latter have
visited the area. Also, the local geographer, Saïd Hachim, is given prominence over members of the scientific monitoring
network, even in situations where they are present during campaigns at sea or interventions with schoolchildren. Similar
observations were made in two other independent studies (Cripps and Souffrin, 2020, which emphasises a general distrust
towards official discourse, and Bedessem et al., 2023, which found confidence in scientists without being able to
determine if this extends to trust in their institutions). In such circumstances, one might question whether there exists a
gap between the representation of journalists and that of local populations, accentuated by a journalistic bias towards
institutions identified as reliable and easily accessible by journalists. In any case, there is a clear need for proximity -
geographical, if not cultural - between sources, given the evident dominance of cited personalities when they are on-site.
With regard to local personalities in particular, their emergence in press coverage occurs later than for others, perhaps
attributable to a search for new sources to compensate for the perceived lack of information on the spot (Fallou et al.,
2019). In any case, the over-representation of individuals compared to institutions raises questions and warrants further
investigation.

This approach offers: i) an overview of the interrelationships between all actors involved in managing this crisis, ii)
highlights specificities linked to the context and media coverage; iii) reflects implicit representations and bonds of trust
between actors, and iv) visualises the network's dynamics over time and how it is disrupted and reorganised after the
occurrence of new events. It aligns with a theoretical background discussed in recent studies, proving relevant for
studying risk and crisis management and governance: Post-ANT (Actor-Network Theory) (e.g. Beck and Kropp, 2011;
Neisser et al., 2014; Bielenia-Grajewska, 2020) and Assemblage Theory (McGowran and Donovan, 2021). This study
aligns with this theoretical background in several ways. First, it involves an empirical examination of interrelations and
associations among actors operating in a simultaneously complex, uncertain, and ambiguous context. Our perspective on
these actors is both relationalist and functionalist: it is the flow of discourse, itself structured by the roles these actors play
in relation to one another, which creates and shapes this network. Moreover, these various actors are networks themselves
and we consider them on an equal footing, including the media used to constitute the dataset. We also manage to depict
the dynamic nature of this network and how it can be affected by the emergence of new actors, relationships,
arrangements, or any other disruptive element, including non-human factors (see section 4.4). Finally this study illustrates
and enhances our understanding of the patterns of ordering. Its originality lies in the fact that we apply this method not
only to study the network of actors in the context of crisis management or governance but also to explore the
representation that certain actors (the media) have of information circulation in this crisis management context. The use
of Actor-Network Theory (ANT) and Assemblage Theory is relevant in this approach, considering their application to



57

discuss the role of mobilisation in communication (Bielenia-Grajewska, 2020) and the compatibility of ANT with media
theory (Couldry, 2008, and Belliger and Krieger, 2015).

**6. Conclusion et perspectives**
This study proposes mapping citation and reference networks of actors identified in 6 daily newspapers as involved in
information circulation during a seismo-volcanic crisis management on Mayotte island from 2018 to 2021. In addition to
providing an overview of the interrelationships between these actors, it offers a dynamic representation of how these
networks evolve over time and how they can be disrupted and reorganised after the occurrence of new events. It also
allows us to identify the common organisation of crisis management as well as some specificities linked to the particular
context of Mayotte or its treatment in the media. This method also reveals implicit representations and trust bonds among
these actors and aligns with results from more detailed analyses. Key findings include an overrepresentation of scientific
actors, both among actors cited in the articles and among actors introduced by a third party, which emphasise the
centrality of scientific discourse in a context where the manifestations of the hazard are mainly visible through their
instruments. It also calls for further research to explain the feeling of "information vacuum" highlighted by Fallou et al.
(2019) among inhabitants despite the abundant communication from scientists and authorities exposed by Devès et al.
(2022a) and their overrepresentation in the mediatic discourse evidenced here. Another important result is the central
representation of individuals, beyond institutions, which suggests varying trust placed in individual versus institutional
discourse. The tension between national and local is evident as also observed in similar contexts in other overseas
departments. From an operational point of view, these results provide keys to identify profiles that have proved decisive
to the flow of information. The fact that these networks are not always interconnected, especially around REVOSIMA
(Mayotte volcanological and seismological monitoring network) which is sometimes completely isolated, also
emphasises the need for diversified information channels to make its circulation more efficient.

764        Here, we use this method to study both networks of actors in a crisis management context and media
representations of the information circulation. However, this method can have numerous other applications, such as
comparing representations between different groups of actors by applying it to several different text corpuses (scientific
or official press releases, media articles, experience reports, etc.) or studying the morphogenesis and the evolution of a
specific actor, by focusing on a particular actor and/or a given period (for instance, to study the scientific cooperations
from the point of view of the media). It is also a first step towards other explorations. For example, it could be refined to
identify the actors cited in support of or in opposition to a given statement. Ultimately, it could also generate datasets for
Artificial Intelligence training to automate the mapping of actors' organisation and information circulation according to a
heterogenous corpus of texts. Finally, this reproducible method can be used for other case studies around the world. It can
also be analysed in other ways to study how scientific information flows from where it is produced to media, explore
tensions between information and entertainment in mediatic discourses and compare mediatic covers of the same events
at different geographical scales (national, regional, local).



59

**Author contributions**

MHD and MLT were responsible for the conceptualization of the study and project administration. MHD and MLT provided a methodology for data collection. MHD conducted the keywords selection and analysis. LLV was responsible for data collection, stockage and investigation. MLT and LLV designed the method used for actors' citation chains identification and encoding. MLT was responsible for the figures and produced the scripts for the analyses reported. MLT, MHD and LLV conducted all analyses. LLV and MLT wrote the original draft of the paper, MD undertook the revision and editing of the final paper. All authors discussed the results and the method.

**Acknowledgements**

This study was carried out within the framework of the MAY'VOLCANO project supported by the IdEx Université Paris Cité, Centre des Politiques de la Terre, ANR-18-IDEX-0001. The authors would like to thank Hugo Pierrot and Geoffrey Robert, whose internship work contributed to verify the completeness of the article database. This publication was financed by the V-CARE project (ANR-18-CE03-0010) and the May'science II project supported by INSU-CNRS.

**Data availability**

As we can not disseminate articles' content because of copyright, an attribute table is made available online on the IPGP dataverse platform (https://dataverse.ipgp.fr/privateurl.xhtml?token=7269cab5-784a-4e85-9653- 64b2784f9f48) with : 1) a link to articles' URLs on newspaper websites, 2) key features such as newspaper name, publication date, authorship, readership and title (see Supplementary Information available on the same IPGP platform for further details). Feel free to contact the authors for more information or access to the whole database. The code used and all appendices are accessible on the same dataverse platform and a valid DOI will be provided for the publication of this article.

**Competing interests**
The authors declare that they have no conflict of interest.

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
