# Peer review of "Tracing the evolving actor network: A social network analysis of"

_EGUsphere, 2024_

## Referee Comment (RC2)

[referee-annotated manuscript omitted]

---

## Author Comment (AC1)

Dear Dr. Mani,

We would like to extend our sincerest thanks for your exceptionally thorough, insightful, and constructive review of our manuscript. We are particularly grateful for your positive assessment of the core contribution and methodology of our work. Your detailed feedback and the provided annotated PDF have been incredibly valuable in guiding our revisions.

We have undertaken a major revision of the manuscript, carefully addressing every point you raised. We believe the paper is now substantially stronger, clearer, and more impactful as a result of your guidance.

Below, we provide a point-by-point response to your comments, detailing the changes we have made.

Sincerely,

The authors

**Point-by-point response**

General comments :

- **Title**: We agree that the original title was not clear enough. As suggested, we have revised it to better reflect the core of our analysis.
    - New Title: *"Tracing the Evolving Actor Network: A Social Network Analysis of the 2018 Mayotte Crisis in the Press"*
- **Visual Timeline**: We propose to quote the following figure which has been already published by one of the co-author in NHESS (Deves et al., 2022). This timeline illustrates perfectly the key phases of the eruption and the disaster response. If the editorial team of NHESS agrees, we can add this figure to the manuscript.

[Figure]

**Figure in Deves *et al.*, 2022 :** Major phases and markers of the response by local and national authorities in charge of risk and crisis management and by scientific experts in charge of monitoring the seismo-volcanic activity in Mayotte. Our period of study extends from the 10th of May 2018 to the 1st of April 2021. SISMAYOTTE, REFMAORE, MAY-MT, and SISMAORE are acronyms of scientific campaigns. SISMAYOTTE was funded by "Tellus Mayotte" and the others by REVOSIMA's institutional partners. The lockdown periods that are shown are those of metropolitan France during the Covid-19 pandemic (note that most of the scientific institutions involved in monitoring are located in metropolitan France). Mayotte endured longer lockdowns in spring 2020 and 2021, but there was no proper lockdown in autumn 2020 (Devès *et al.*, 2022).

- **Use of 'Corpus':** Thank you for this linguistic advice. To ensure clarity and use more common terminology, we have replaced "corpus" with "selection of articles" throughout the manuscript. This sub-section is now titled "Selection of articles"
- **Stylistic Preference (avoiding "we"/"our")**: We appreciate this stylistic guidance. The manuscript has been revised to adopt a more objective, third-person tone, with first-person pronouns removed wherever possible.
- **Actor Context:** This point is important for contextualising the results. However, a detailed explanation would disrupt the flow of the article. Instead, we propose to refer (in section 3.2.1.) to a published article by one of the co-authors (Devès et al., 2022a), which offers a more comprehensive account of the main actors and actor categories listed in Table 4, outlining their expected roles, responsibilities, and capacities during the crisis, thereby providing essential context for the subsequent network analysis.
- **Consistency (Figures, Tables, Dates):** Thank you for spotting these inconsistencies. We have performed a careful check of the entire manuscript to ensure uniform formatting for all figure/table references (using "Figure" and "Table"

consistently) and for all dates (using the "Day Month Year" format, e.g., the 5th of October 2018).

- **Results vs. Discussion:** We had inadvertently included interpretive statements in the Results section. We have now moved these sentences (formerly lines 350-351, lines 374-379, lines 416-419) to discussion in order to maintain a clear and logical structure.
- **Name Consistency** (Said Hachim): This oversight has been corrected. The name "Saïd Hachim" is now used consistently throughout the text and in all relevant figures.
* * *
Specific Comments

We thank you again for the line-by-line suggestions in the attached PDF, which we have implemented almost entirely. Below we address the main specific points.

**Introduction:**

- **Literature Review Style:** We have revised the introduction to integrate the literature more fluidly. The parenthetical explanations have been removed, and the relevance of cited works is now woven directly into the narrative of the text.
- **Recent References:** This is an important point, thank you for highlighting it. We have revised our literature review and incorporated several recent, key studies on crisis communication and media analysis in the opening paragraphs, in order to better frame our work within the current state-of-the-art.
- **Sentence Rephrasing (L60, L88, etc.):** All sentences you highlighted have been rephrased for clarity and grammatical accuracy. For instance, the sentence on L88 now reads "as the approach allows for: i) gaining insights into the actual organisation of actors by providing a comprehensive view of all cited actors and their interactions, allowing the detection of communities (e.g. Park et al., 2015 and Williams et al., 2015)". We have also clarified the use of the expression "blur messages", which we quote from previous articles : "The way journalists tend to juxtapose the accounts of heterogeneous sources, while important for depicting a variety of viewpoints, has been shown to "blur" messages (e.g. Lejeune, 2005, Léglise and Garric, 2012, and Devès et al., 2022a), thereby reducing their clarity."

**Case Study Description:**

- **Formulations :** We have implemented all the suggested changes regarding titles, phrasing (e.g. "monitoring network," "earthquakes"), and sentence structure (e.g. in L113, L258-262, L206, …). Speculative statements have been removed (L185). L117-125 have been restructured as follows : "From a scientific perspective, uncertainties were exceptionally high, especially in the first months of the seismic crisis, due to scarce knowledge of the geodynamical context in the area and a poor monitoring network (Saurel et al., 2021; Bertil et al., 2021; Feuillet et al., 2021). The recorded signals were poorly constrained in terms of location and magnitude and

remained difficult to explain in this region. The volcanic hypothesis to explain the origin of the seismic activity did not emerge until several months later, in October 2018, and was not confirmed until May 2019. This made public communication particularly difficult and led to the development of a "technicalist bias", with frequent, but minimalist communication from institutions that did little to help the population gain situational awareness (Devès et al., 2022a)." L127 has been revised to "Another sign of this still ongoing activity is the detection at 10 to 15 km off the coast of acoustic plumes associated with geochemical anomalies (22 sites observed in July 2022, MAYOBS 23)", in order to highlight the fact that this activity is still ongoing.

- **Title of Section 2. :** This section was renamed "The 2018 Mayotte seismo-volcanic crisis"

**Method**

- **Building on previous studies :** In our method L165-168, we quote two studies (Deves *et al. 2022a ; Devès et al., 2022b)* that were indeed not well summarised. We propose the following reformulation to precise the importance of these studies for the method we use here and better build upon their works : "This study builds on two previous studies. The first, by Devès et al. (2022a), focused on public information processes and identified shortcomings in both scientific and state institutional communication. The second study, by Devès et al. (2023), illustrated how newspapers implicitly reproduce asymmetrical power relationships between actors (e.g., local versus national authorities, experts versus lay public). Building on these findings, this study aims to identify and compare the presence of actors according to their role in the risk reduction network, the geographical scope of newspapers (local versus regional versus national), and whether there are significant differences between newspapers."

- **Double-reading method:** As our original formulation was not entirely adequate, since this approach cannot strictly be described as a 'method', we have clarified our process with the following sentence: "To study the press coverage of different categories of "actors", a double-reading process was employed. Two researchers independently reviewed the articles to identify each actor or group of actors mentioned, even when they were identified by professional status, by nicknames, etc."

- **Figure 1 :** This figure has been expanded for better readability and is now labelled as Figure 2.

- **Table 2:** We have expanded the caption of Table 2 to better explain the logic of the adjacency matrix, including a precise definition of the rules applied.

- **Added lines :** We have added line breaks where suggested (e.g. L182, L234, L343)

- **Methodological Definitions** (e.g., Louvain clustering): We thank the reviewer for their careful reading and insightful comments. We would like to clarify that the mention of global network indicators and the Louvain clustering method, which may have appeared in earlier drafts, was already removed from the version submitted for review. However, we recognize that the wording of the paragraph may have caused confusion, and we sincerely apologize for that. Following the reviewer's helpful

suggestion, we have revised the paragraph to explicitly state the use of the *igraph* package in R for computing node-level centrality indices. The updated paragraph now reads: *"We study the system of actors depicted by the network of citations to better understand the relationships between individual actors, actor categories and their evolutions. This is accomplished using node-level centrality indices, including in-degree, out-degree, and betweenness centrality, computed with the igraph package in R. Network diagrams plot citation links with arrows, and the size of the nodes, as well as the font size of generic names, are weighted according to their degree, representing the number of direct connections each node has within the citation network. Unidentified actors are removed from the graphs to avoid generating false co-citation relationship structures."*

**Results:**

- **Figure Quality (Figures 1, 3, 5-11):** We acknowledge that the figures were not legible. All figures have been completely recreated at a higher resolution, with larger font sizes, improved layouts, and clearer captions when technically possible.
- **Figure 5 (The "unreadable" network)**: This is a critical point that was also raised by Reviewer 1. As mentioned in the introductory remarks of our response to Reviewer 1, we have taken your suggestions into account in revising this figure. Specifically, we reduced the font size and repositioned labels where overlaps occurred. While we made these adjustments to improve readability, we would also like to emphasize that the visual density of the network is intentional and represents a key result of our analysis. The figure's initial "illegibility" is meant to illustrate the multitude of actors and the unstructured, "hairball" nature of the communication network during the crisis. This visual evidence supports our argument about the challenges of communication and coordination.
- **Figure 3 :** This figure has been relocalized correctly.  Figure 3 – can you find a way to plot this so we can see the nuance of the data in the bottom left more clearly? E.g. break the X axis
- **Figure 6** : We've moved the labels that touch each other to make them easier to read. However, we are awaiting discussions with the editors at the time of the final layout to decide on the size of this figure.
- **Actor Clarification (Prefect, Rectorate)**: Thank you for highlighting this ambiguity for a non-French audience. We have clarified these roles in the text (e.g., " which is the body representing and implementing government policy at the local level", "the Prefect (the State's representative in the department" and "the rectorate, the decentralized government department for education, and teachers"). However, we kept this denomination, since there is no good equivalent in the Anglo-Saxon systems.
- **Other populations :** This formulation refers to all populations living outside of Mayotte and France (mostly non Mahorese). A detailed definition of each category is provided in the Supplementary Information.

**Discussion and Conclusion:**

- **Temporal Perspective in Discussion:** This is the most crucial intellectual point of your review, for which we are sincerely grateful. You are absolutely right in noting that our initial discussion did not fully engage with the temporal dynamics revealed by our findings. In response, we have added a paragraph to the discussion section that addresses the observed shifts in actor relationships and prominence across the different phases of the crisis. This new paragraph focuses in particular on the evolving position of REVOSIMA over time, in relation to the local figure Saïd Hachim, who occupies a notably central position within the network.

- **Associated press:** We included a short passage reminding the role of the associated press and its use in international and national press practices : "with local journalists relying more on local actors and national journalists adopting alternative strategies, such as using social networks (e.g., X and Facebook for national dailies like *Le Figaro* and *Le Monde*) or relying on news agencies like AFP (Agence France-Presse), or their local counterparts (Lecheler and Kruikemeier, 2016), directly, or indirectly, via associated press (e.g. Reuters) who provide articles for international media based on local reporting."

- **Interest of two-way communication :** We added a sentence explaining why two-way communication could be important and beneficial ("Two-way communication has the potential to facilitate cooperative decision-making (Renn, 2009), to draw upon local knowledge in order to improve understanding of field-level dynamics (Lindell et al., 2006), and, in turn, to strengthen public engagement and contribute to the development of trust (Leiss, 1996; Renn, 2009)"

- **Mention Sentiment (Positive/Negative):** This is an excellent point and a fascinating avenue for research. While a full sentiment analysis is beyond the scope of the current study, we agree it is a key perspective. We have acknowledged this in our discussion and discussed our results with other studies using qualitative analyses methods on the same case in order to overcome this limitation. We also mention this in the "Conclusion and perspectives" section as a perspective for further work.

- **Analysis versus planned crisis communication for Mayotte**: At the time of writing, no official volcanic crisis communication strategy has yet been established for Mayotte. However, based on what exists in other departments, such a strategy would likely prioritise top-down, institutional communication channels. In contrast, our findings highlight the central role played by identified individuals—particularly local figures—as key sources within the communication network.

- **Future uses of this method:** These considerations have been retained in its final paragraph of the 'Conclusion and Perspectives' section.

- **Limitations of the Study**: Several limitations of this study have been identified and presented in the text, including the likely gap between the perspectives of journalists and those of local populations, the absence of clear indications regarding each actor's stance in relation to specific statements or other actors, and the decision to focus solely on six French-language daily newspapers. In the 'Conclusion and Perspectives' section, we outline several points for consideration for further research to address these limitations. These include refining the methodology to identify whether actors are cited in support of or in opposition to particular statements, examining how actors perceive one another, expanding the selection of articles, and exploring the use of AI to automatically map actor networks and information flows

from larger and more diverse corpora. We also suggested conducting sociological field studies to investigate local populations' perceptions.

- **Conclusions and Perspectives:** We have corrected the title of the section to "Conclusion and Perspectives".
- The conclusion has been revised to be more focused and impactful and to only mention points that have been developed above in the discussion.
* * *
We believe these extensive revisions have substantially improved the manuscript's clarity, rigor, and overall contribution. We are very grateful once again for your expert guidance and for the opportunity to strengthen our work.

Sincerely,

The Authors

---

## Author Comment (AC3)

Dear Reviewer 1,

We would like to express our sincere thanks for your positive feedback and for the constructive comments you provided on our manuscript. We were pleased to read that you found the analysis both well-discussed and fair. We have carefully addressed all the editorial and technical points you raised.

In response to your comments, the entire manuscript has undergone thorough proofreading to improve English grammar and syntax. Furthermore, all the figures you mentioned (Figures 1, 5-11) have been remade in higher resolution with enlarged font sizes, wherever technically feasible, to enhance readability.

With regard to your specific comment on Figure 5, your observation aligned witha similar concern raised by Reviewer 2, and we have taken both sets of feedback into account in revising this Figure 5. The font size has been reduced slightly and label positions have been adjusted to minimise overlap. We agree that the network is visually dense. This visual complexity is intentional and  constitutes a key finding of our analysis. The initial 'illegibility' of the figure is meant to reflect the multitude of actors and the unstructured, "hairball" nature of the communication network during the early stages of the crisis. This visual evidence supports our argument about the challenges of communication and coordination. This visual feature reinforces our argument concerning the difficulties in communication and coordination. To clarify this intention, we have added an explanatory sentence to the figure caption and have highlighted the main actors more clearly within the diagram.

Please find our detailed, point-by-point responses to your suggestions below.

Sincerely,

The Authors

**Point-by-point response**

- Line 114, "eruptive activity at sea": please consider revising to "submarine eruptive activity". => Done

- Line 114, "a newly born": Please use a reference to confirm that it is indeed new and that it is not a preexisting that was simply discovered in 2019.  => Done (Feuillet et al., 2021).

- Line 115, "uncertainties were really high": uncertainties with respect to what? The location of the volcano? The origin of the felt earthquakes? The duration of the crisis? Please specify if there is a size or a magnitude (and units of measurement or magnitude units) used to describe the uncertainty. => Uncertainties were both instrumental and epistemic, as the geodynamics of this region were not well constrained and the sensor network was poor. It was therefore difficult to determine both the origin of this seismic activity and the duration of the crisis. We have added a few elements to the text to clarify this :

"From a scientific perspective, uncertainties were exceptionally high, especially in the first months of the seismic crisis, due to scarce knowledge of the geodynamical context in the area and a poor monitoring network (Saurel et al., 2021; Bertil et al., 2021; Feuillet et al., 2021). The recorded signals were poorly constrained in terms of location and magnitude and remained difficult to explain in this region. The volcanic hypothesis to explain the origin of the seismic activity did not emerge until several months later, in October 2018, and was not confirmed until May 2019."

- Line 117, "poor instrumental network": Poor in what sense? Not dense enough? Bad resolution in time, bad bit rate, something else? => It mainly concerned the spatial distribution and density of the sensors. The type of sensor was also at issue, since there were no broadband frequency stations on site during the first few weeks of the seismic crisis. We have added a few details to characterise this situation in the main text of the article. See corrected paragraph above.

- Line 117, "instrumental network": revise to "network of sensors" => Done

- Line 119, "to appraise the situation": Please consider revising to "to have situational awareness". => Done

- Line 126, "seisms": replace with "earthquakes". => Done

- Line 127, "km from the coast": replace with "off the coast". => Done

- Lines 128-129, "on Petite Terre island": replace with "on the Petite Terre island, Mayotte's second-largest island, east of the largest island and closer to the Fani Maore volcano". => Done

- Line 192, Figure 1: The image resolution is too low, which makes the text hard to read, which is already hard to read because of the small font size. Please consider improving the figure while taking into account any editorial requirements. => Resolution was improved, and font size was enlarged.

- Line 207, "a double-reading method": Please give a short description of this method, perhaps a reference too. => Our formulation is actually not adequate here since this way of proceeding can not be labelled as a method. Thus, we propose a reformulation : "To study the press coverage of different categories of "actors", a double-reading process was employed. Two researchers independently reviewed the articles to identify each actor or group of actors mentioned, even when they were identified by professional status, by nicknames, etc."

- Line 214, "20190507_JDM_001": It seems that this needs to be deleted. => This was an example among articles selected in our corpus. We have reworded the sentence to quote the reference correctly :

  "Nathalie Feuillet, a researcher, was wrongly affiliated with IFREMER in some articles, such as in an article published in the Journal de Mayotte on the 7th of May 2019"

- Line 216, "its exact denomination(s)": Incorrect English; replace with "their exact denominations".=> Done

- Line 225, Table 1; Line 438, Figure 5: "sismo-volcanic": typos; replace with "seismo-volcanic" => Done

- Line 258, "Louvain clustering method"; Line 260, "network diagrams plotting citation links" : cite any code or software used to apply this method and to plot the diagrams or declare that you used an in-house code (specify the programming language you used in this case, and cite it). => We thank the reviewer for their careful reading and insightful comments. We would like to clarify that the mention of global network indicators and the Louvain clustering method, which may have appeared in earlier drafts, was already removed from the version submitted for review. However, we recognize that the wording of the paragraph may have caused confusion, and we sincerely apologize for that. Following the reviewer's helpful suggestion, we have revised the paragraph to explicitly state the use of the igraph package in R for computing node-level centrality indices. The updated paragraph now reads:

  "We study the system of actors depicted by the network of citations to better understand the relationships between individual actors, actor categories and their evolutions. This is accomplished using node-level centrality indices, including in-degree, out-degree, and betweenness centrality, computed with the igraph package in R. Network diagrams plot citation links with arrows, and the size of the nodes, as well as the font size of generic names, are weighted according to their degree, representing the number of direct connections each node has within the citation network. Unidentified actors are removed from the graphs to avoid generating false co-citation relationship structures."

- Lines 253; 309, : replace "vs" with "versus" throughout the manuscript, e.g.: "speech vs simple" revise to "speech versus simple"; "mention vs indirect mention" to "mention versus indirect mention", etc.. => Done

- Line 367, "achieving a PhD in geography at Paul Valéry Montpellier 3 University": Incorrect English, revise to "holding a PhD in geography from the Paul Valéry Montpellier 3 University". => He is actually a PhD candidate. We have made the correction :

  "Saïd Hachim, a geographer from Mayotte (Mahoran) who works at the Departmental Council of Mayotte and is also a PhD candidate in geography at Paul Valéry Montpellier 3 University in mainland France"

- Line 374, "BRGM (French geological survey BRGM)": revise to "BRGM (French geological survey)". => Done

- Line 378, "representatives, ect)": replace with "representatives, etc.)".=> Done

- Lines 395-396, "oceanographic campaigns (MayObs 1 and 2)": please add a reference. => Done

- Line 398, Figure 3: Please consider using colorblind friendly colors. => Thank you for this important remark. In response, we revised the color palette of Figure 3 using a pair of contrasting colors — #E66100 and #5D3A9B — which maintain their contrast for individuals with color vision deficiencies.

- Lines 402-403, "presented in green", "presented in orange": revise to "annotated with green/orange labels". => Done

- Line 417, "Groupe d'Intervention Macrosismique": please give a short description of what this is, and a reference as well. => We reformulated as followed and moved this paragraph into the discussion :

  "The municipality of Chirongui stands out here, probably because it hosted the delegation of specialists in civil security and natural risks dispatched by two ministries (Ministry of Ecology and Ministry of the Interior) (Journal de Mayotte, the 6th of June 2018). This event served as an entry point for the presentation of the group in the local press. Furthermore, the mayor at the time remained in office from 2008 to 2020, and her team appears to have been particularly active and well integrated into the local community."

- Line 436, Figure 5: Not very helpful, too dense network, hard to see the connections. => We refer you to the introductory remarks in our response letter.

- Line 454, "mainland France and": add a serial comma "mainland France, and". => Done

- Line 456, "Macroseismic Intervention Group": Here it is mentioned in English; elsewhere the manuscript used its French name. Please use the same names consistently through the text. => Done

- Line 489, "VLP earthquake": Please write it out in full: "very-long-period earthquake". => Done

- Line 491, "Twitter network": Delete the space: "Twitter network". => Done

- Line 586, "several limitations in our study": Please consider briefly discussing all the limitations of this study that you are aware of. => Actually, we detailed those limitations in the same paragraph. We agree that this formulation is not appropriate as it suggests more limitations so we propose the following :

  "Despite the focus on newspaper representations rather than those among populations, and the use of articles from six non-specialist French-language newspapers, this study provides a comprehensive insight into media narratives during the seismic-volcanic crisis in Mayotte from May 2018 to May 2021."

- Line 628, "relatively minimal material and human damage (only three lightly injured and cracked buildings in Mayotte)": Please revise to "the light building damage (cracks), and the small number of people affected (three lightly injured)." => Done

- Line 636, "However, here despite": please revise to "However, here, despite" => Done

---

## Author Response (AR2)

**Authors' answer to the editor's review**

Dear Editor,

Thank you for your positive evaluation and for your two helpful remarks.

Following your suggestions, we have removed the expression 'could be improved', which we had inadvertently left at the end of line 761. We have also moved the paragraph outlining the aim of the studies and describing the two previous studies it builds upon, from the beginning of Section 3.1. to the end of the Introduction.

We just uploaded two versions of the revised manuscript to the web platform: one with tracked changes, and one with the changes accepted.

Wishing you a pleasant summer.

Best regards,

Louise Le Vagueresse and co-authors.

---

## Author Response (AR3)

**Authors' answer to the editor's last review**

Dear Editor,

Thank you for accepting to publish this manuscript.

In this very last version, we have sed full first names for all authors, as suggested by the editor board.

Best regards,

Louise Le Vagueresse and co-authors.